# OASIS: One-Shot Federated Graph Learning via Wasserstein Assisted Knowledge Integration

**Guancheng Wan**[1†], **Jiaru Qian**[1†], **Wenke Huang**[1†], **Qilin Xu**[1], **Xianda Guo**[1],
**Boheng Li**[2], **Guibin Zhang**[3], **Bo Du**[1], **Mang Ye**[1*]
[1]School of Computer Science, Wuhan University    [2]NTU    [3]NUS
{guanchengwan, yemang}@whu.edu.cn

## Abstract

Federated Graph Learning (FGL) offers a promising framework for collaboratively training Graph Neural Networks (GNNs) while preserving data privacy. In resource-constrained environments, One-shot Federated Learning (OFL) emerges as an effective solution by limiting communication to a single round. Current OFL approaches employing generative models have attracted considerable attention; however, they face unresolved challenges: these methods are primarily designed for traditional image data and fail to capture the fine-grained structural information of local graph data. Consequently, they struggle to integrate the intricate correlations necessary and transfer subtle structural insights from each client to the global model. To address these issues, we introduce `OASIS`, an innovative one-shot FGL framework. In `OASIS`, we propose a Synergy Graph Synthesizer designed to generate informative synthetic graphs and introduce a Topological Codebook to construct a structural latent space. Moreover, we propose the Wasserstein-Enhanced Semantic Affinity Distillation (WESAD) to incorporate rich inter-class relationships and the Wasserstein-Driven Structural Relation Distillation (WDSRD) to facilitate the effective transfer of structural knowledge from the Topological Codebook. Extensive experiments on real-world tasks demonstrate the superior performance and generalization capability of `OASIS`, with an average improvement of 15.81% over the baseline. The code is available for anonymous access at https://github.com/JiaruQian/OASIS.

## 1 Introduction

Federated Learning (FL) [38, 19, 20] enables decentralized model training, allowing collaboration across clients while preserving privacy. Many real-world applications involve non-Euclidean data structures, such as *graphs*, where entities are interconnected through complex relationships. These graph-structured data are common in various domains, including biological networks [69], urban mobility systems [68], and online social platforms [47]. To learn from such structures from multiple participants, Graph Neural Networks (GNNs) [25, 58] have been integrated with FL, leading to Federated Graph Learning (FGL) [9, 31]. This approach combines both paradigms, ensuring privacy while enabling efficient learning on distributed graph data.

Although most research in FGL has focused on personalized learning, where each client has a tailored model [3, 73, 30], there is growing demand for a global model that can generalize across diverse graph data from multiple clients [52]. This is especially important in scenarios with limited data or where consistency between clients is necessary, such as in medical data analysis across

---

[†] Equal Contribution.
[*] Corresponding Author.

39th Conference on Neural Information Processing Systems (NeurIPS 2025).

hospitals or regional traffic network predictions [57, 7]. However, training a global model in traditional FGL requires multiple rounds of communication, which can be burdensome for edge devices with limited bandwidth and computational power [13, 2]. To address these challenges, *One-shot Federated Learning (OFL)* has been proposed, restricting communication to a single round [12, 34], thus reducing communication costs and potentially enhancing privacy [62]. However, existing OFL methods are primarily designed for traditional data types like images and do not tackle the unique challenges of graph-structured data. We define the task of **Generalizable One-Shot Federated Graph Learning** as developing a global model capable of generalizing well for graphs distributed across multiple clients using only one round of communication. A key research question is:

*How to design a generalizable OFL framework, specifically tailored for graph-structured data?*

Several OFL approaches have proposed using public datasets to train a global model. However, this approach may be impractical because of the limited availability of suitable datasets or due to stringent data sensitivity constraints [46, 36]. Consequently, generative methodologies have emerged as compelling alternatives. For instance, DENSE [66] employs a strategy in which each client trains a generator to produce synthetic data that reflects its distribution. In contrast, FedDEO [61] utilizes diffusion models as generators, leveraging client-trained descriptors to guide the server in training a global model. Nevertheless, these methods are primarily designed for image data and do not capture the fine-grained structural information inherent in nodes, particularly substructure variations such as connectivity patterns and neighbor distributions, as illustrated in Figure 2. The direct application of these generators fails to encapsulate the spatial intricacies of graphs, thereby limiting their effectiveness. This limitation raises a critical question: **I)** *How can client generators better capture the fine-grained structural knowledge of local graphs?*

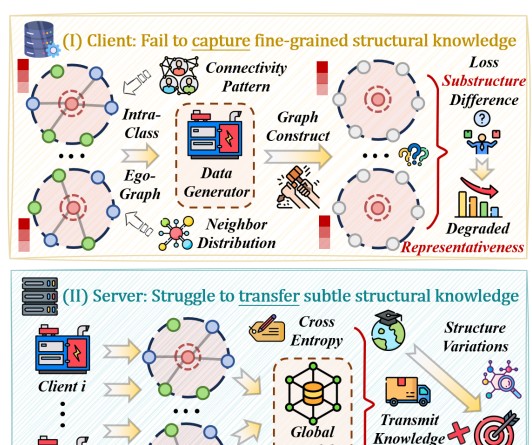

Figure 1: **Problem Illustration**. For current one-shot FGL scenarios: **I)** at the client stage, existing generative methods fail to capture the fine-grained structural knowledge of local graphs; **II)** at the server stage, conventional KD techniques are unable to effectively transfer the subtle structural characteristics from diverse clients, thereby hindering the global model generalization.

After training the local generator, several methods have leveraged knowledge distillation (KD) on the server to train the global model [16, 41]. For example, DENSE first ensembles local models and then distills the resulting ensemble, while FedCVAE-KD [15] employs KD from local decoders to train the server decoder. However, these distillation methods depend excessively on class-related semantic signals when aggregating client knowledge, and the class space may not be sufficiently expressive to represent the diverse local graph structures of nodes. Consequently, they fail to capture the intricate correlations and struggle to transfer the clients' structural awareness to the final global model. This observation raises an important follow-up question: **II)** *How can we effectively transfer subtle structural knowledge during server distillation?*

To address the aforementioned challenges, this paper presents the first systematic study on the problem of generalizable one-shot FGL. We introduce the **OASIS** framework: **O**ne-Shot Federated Gr**A**ph Learning via Wasserstein As**SIS**ted Knowledge Integration. To address problem **I)**, we introduce the Synergy Graph Synthesizer, which is designed to generate informative synthetic graphs. This approach is further refined by incorporating alignment constraints based on the Fused Gromov-Wasserstein distance, which effectively harmonizes the reconstruction of both features and structures. More importantly, inspired by [55, 64], we construct a novel structural latent space tailored for graph data. This enables the learning of a structure-aware tokenizer that encodes each node along with its substructure as a discrete code, thereby capturing its spatial characteristics. By utilizing the established *Topological Codebook*, we can precisely characterize subtle structural variations, thereby enabling the meticulous integration of fine-grained structural knowledge from client graphs. On the server side, to address issue **II)**, we first propose the Wasserstein-Enhanced Semantic Affinity Distillation, which models the rich inter-class relationships by distilling logits from the teacher local model to the

student global model, while learning more general structural differences. At the intra-class level, we introduce the Wasserstein-Driven Structural Relation Distillation, which dynamically measures the distance between samples and the discrete codes derived from the *Topological Codebook*, thereby facilitating the sophisticated transfer of subtle structural knowledge. Our principal contributions are summarized as follows:

❶ *Problem Identification.* We identify the key challenges in generalizable one-shot FGL: how to effectively capture fine-grained structural knowledge of local graphs, and then how to transfer this knowledge during distillation to enable global model generalization.

❷ *Practical Solution.* We propose a novel approach that integrates the Synergy Graph Synthesizer with Topological Codebook capturing subtle structural variations, then we introduce hierarchical Wasserstein-based distillation to skillfully transfer structural knowledge on server.

❸ *Experimental Validation.* We examine `OASIS` through extensive experiments on various graph datasets, demonstrating superior generalization ability for one-shot FGL.

## 2 Preliminaries

### 2.1 Notations

**Graph Neural Networks.** Consider a graph $\mathcal{G} = (\mathcal{V}, \mathcal{E})$, where $\mathcal{V}$ denotes the set of $N$ nodes, and $\mathcal{E}$ represents the edges. Each node $v_i$ is associated with an $F$-dimensional feature vector $x_i$, and these vectors collectively form the feature matrix $\mathbf{X} = \{x_1, x_2, \ldots, x_N\}^\top$. The structure of $\mathcal{G}$ is encoded in the adjacency matrix $\mathbf{A} \in \mathbb{R}^{N \times N}$, where the entries are defined such that $\mathbf{A}(i, j) = 1$ if nodes $i$ and $j$ are connected, and $\mathbf{A}(i, j) = 0$ otherwise. Graph Neural Networks (GNNs) iteratively build node representations by aggregating features from neighboring nodes and then applying an update function. Specifically, the representation $h_i^{l+1}$ of node $v_i$ at layer $l + 1$ is computed as:

$$h_i^{l+1} = \text{Update}\Big(h_i^l, \text{Aggregate}\big(\{h_j^l : v_j \in \mathcal{N}(v_i)\}\big)\Big), \tag{1}$$

where $h_i^l$ is the current representation at layer $l$, $\mathcal{N}(v_i)$ denotes the set of neighbors of node $v_i$, Aggregate$(\cdot)$ combines the neighbor features, and Update$(\cdot, \cdot)$ updates the node's representation. Initially, $h_i^0 = x_i$.

**Problem Formulation.** In a One-shot Federated Graph Learning framework, a central server coordinates $K$ clients (collectively denoted as $\mathcal{C}$, where each client is indexed by $k$). Each client $k$ holds its own graph $\mathcal{G}^k = (\mathcal{V}^k, \mathcal{E}^k)$ with the corresponding adjacency matrix $\mathbf{A}^k$. For every node $v_i \in \mathcal{V}^k$, there exists an associated feature vector $x_i^k$ and a label $y_i^k$, if it is available. Within this setup, each client trains a model $\mathcal{F}_{\theta^k}$ parameterized by $\theta^k$ and then transmits them to the server. The objective is to minimize:

$$\min_\theta \sum_{k=1}^K \frac{N^k}{\mathbb{N}} L^k(\phi). \tag{2}$$

This is computed as the weighted sum of the $K$ local nodes $N^k$, where $\mathbb{N}$ denotes the total number of samples across all clients. The local objective $L^k(\cdot)$ is typically defined as the expected error over all nodes from the local graph $\mathcal{G}^k$, with the global model parameterized by $\phi$. Unlike conventional FGL, one-shot FGL restricts the exchange to a *single communication round*.

### 2.2 Motivation

This paper systematically explores the challenge of maximizing the generalization ability of the global model in one-shot FGL. Traditional OFL approaches that rely on generative models such as Variational Autoencoders or Generative Adversarial Networks—originally designed for image-based data—fail to deliver competitive performance on graphs, as demonstrated in Sec. 4. The primary issue is their inability to effectively capture the spatial and relational dynamics between nodes. These models emphasize coarse, class-based semantic signals while overlooking the underlying substructural variations among nodes. To better illustrate this, we present the t-SNE visualization of the learned graph representation space after training the teacher GNN and the Topological Codebook. (Detailed methods are illustrated in Sec. 3.3.)We select three categories with their representative nodes and corresponding two-hop substructures. Results are shown in Figure 2, where nodes within

the same class can exhibit significantly different connectivity patterns and neighbor distributions. Consequently, the key objective in the local training stage of one-shot FGL is to *identify a knowledge*

*repository that optimally encapsulates both local semantics and fine-grained structural insights.* We formally define this pursuit as follows:

$$\mathcal{R}_{\varphi}^{\star} = \arg\min_{\varphi} \mathbb{E}_{\substack{\mathcal{G}^* \sim \mathcal{R}_{\varphi}(\mathcal{G}^\star | \mathbf{X}, \mathbf{A}), \\ (\mathbf{X}, \mathbf{A}) \sim \mathcal{G}}} \mathbb{Q}(\mathcal{G}^*; \mathcal{G}), \quad (3)$$

Here, $\mathcal{R}_{\varphi}^{\star}$ denotes the optimal knowledge repository obtained by minimizing the expected distance $\mathbb{Q}$ between the generated graph $\mathcal{G}^*$ and the original graph $\mathcal{G}$. In this formulation, $\mathcal{R}_{\varphi}$ is a parameterized function that produces $\mathcal{G}^{\star}$ given the node features $\mathbf{X}$ and the adjacency matrix $\mathbf{A}$. After obtaining the powerful knowledge repository $\mathcal{R}_{\varphi}^{\star}$, our goal is to *integrate these granular insights into the global model $\mathcal{F}_{\phi}$ during the server distillation phase without erosion of knowledge*:

$$\mathcal{F}_{\phi}^{\star} = \arg\min_{\phi} \sum_{k=1}^{K} \mathbb{E}_{\hat{\mathcal{G}}^k \sim \mathcal{R}_{\varphi}^k} \mathbb{T}\Big(\mathcal{F}_{\phi}(\hat{\mathcal{G}}^k); \mathcal{F}_{\theta^k}(\hat{\mathcal{G}}^k)\Big).$$
$$(4)$$

For each client $k$, $\mathcal{R}_{\varphi}^k$ represents a specialized knowledge repository. The global model $\mathcal{F}_{\phi}$ is trained to align its outputs with those of the corresponding client models, while $\mathbb{T}$ quantifies the knowledge difference between them. Minimizing this expected transfer loss ensures that the ultimate global model $\mathcal{F}_{\phi}^{\star}$ effectively assimilates the granular information encapsulated within the knowledge repositories. Based on these discussions, we present the principle for designing the ideal Generalizable one-shot FGL pipeline:

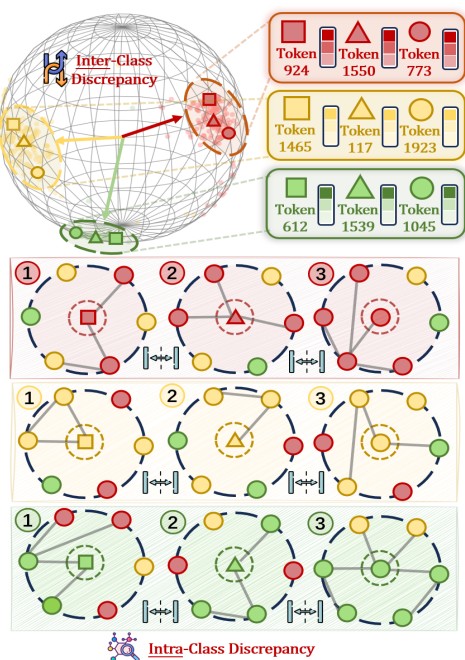

Figure 2: **Motivation.** In the t-SNE visualization, nodes from different categories (colored in red, yellow and green ) are scattered, representing *inter-class discrepancy*. However, nodes within the same category are also mapped to **different** topological codebook tokens (We visualized three for each color), indicating the *intra-class discrepancy* in terms of connectivity patterns and neighbor distributions. Please refer to Sec. 2.2 for details.

> **Generalizable One-shot FGL Design Principle**: *Communication Efficiency: Achieve robust generalization via a single, streamlined communication round, thereby reducing overhead without compromising performance.* **Knowledge Extraction**: *Precisely capture both local semantic signals and fine-grained structural variations from graphs.* **Knowledge Integration**: *Seamlessly integrate these granular insights into the global model, ensuring minimal knowledge loss during the server distillation.*

In following sections, we will elaborate on how OASIS adheres to these principles, encapsulates fine-grained knowledge and then effectively distills it to the global model.

# 3 Methodology

## 3.1 Framework Overview

In this section, we present an overview of OASIS. On the client side, we perform Local Graph Knowledge Extraction, where a synergy graph synthesizer and a topological codebook are constructed to capture fine-grained structural knowledge. After the one-shot communication, we employ Wasserstein-Enhanced Semantic Affinity Distillation and Wasserstein-Driven Structural Relation Distillation to transfer intricate topological knowledge during server distillation. The framework illustration is provided in Figure 3.

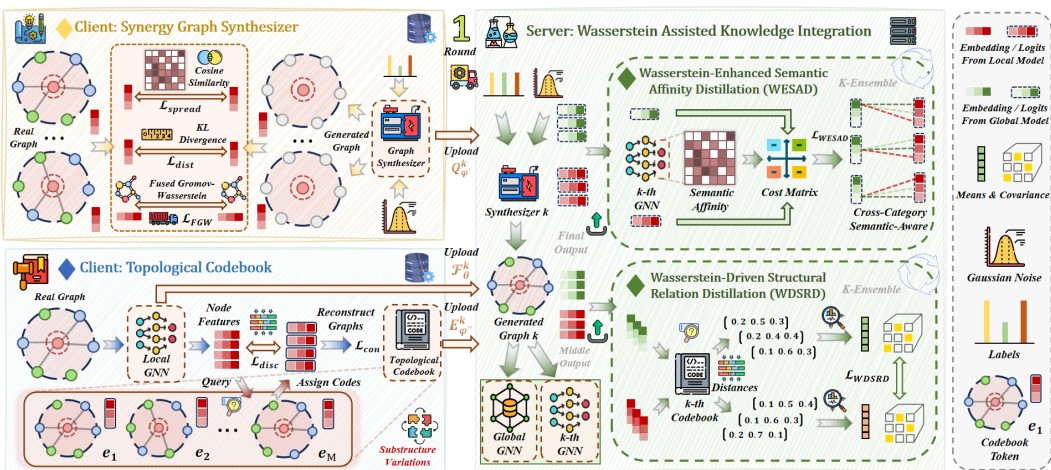

Figure 3: Architecture illustration of `OASIS`. (a) ***The left part*** shows the training process of Synergy Graph Synthesizer and Topological Codebook on the client side. (b) ***The right part*** presents the knowledge integration part on the server side. The communication round is limited to one, with codebooks and parameters of the local GNN and Synthesizer uploaded. The far right section displays the legend. Zoom in for details.

## 3.2 Local Graph Knowledge Extraction

**Synergy Graph Synthesizer.** To construct high-fidelity graphs, we introduce a novel Synergy Graph Synthesizer $\mathcal{Q}_{\varphi'}$ as part of the knowledge repository $\mathcal{R}_{\varphi}$ (omitting $k$ for brevity), which reconstructs both node features and graph topology in a structured manner. Given a graph $\mathcal{G} = (\mathcal{V}, \mathcal{E})$, for each node $v_i \in \mathcal{V}$, a synthetic feature vector is generated by mapping Gaussian noise $\epsilon$ (sampled from a standard Gaussian distribution) and the corresponding label $y_i$ through the generator: $\hat{x}_i = \mathcal{Q}_{\varphi'}(\epsilon; y_i)$, where $\mathcal{Q}_{\varphi'}$ is parameterized by $\varphi'$. This process constructs the synthetic feature matrix $\hat{\mathbf{X}} = [\hat{x}_1, \hat{x}_2, \ldots, \hat{x}_N]^\top \in \mathbb{R}^{N \times F}$. Next, an activated similarity matrix is computed as $\mathbf{H} = \sigma\left(\hat{\mathbf{X}}\hat{\mathbf{X}}^\top\right)$, where $\sigma$ denotes the sigmoid function. By applying a $K$-Nearest Neighbors strategy to each row of $\mathbf{H}$, we obtain the synthetic adjacency matrix $\hat{\mathbf{A}}$:

$$\hat{\mathbf{A}}(i,j) = \begin{cases} 1, & \text{if } j \in \text{TopK}\big(\mathbf{H}(i)\big), \\ 0, & \text{otherwise.} \end{cases} \tag{5}$$

Thus, the synthesized graph is denoted as $\hat{\mathcal{G}} = (\hat{\mathbf{X}}, \hat{\mathbf{A}})$ while preserving the original labels $y = [y_1, y_2, \ldots, y_N]^\top$. We adopt a transductive setting, where some nodes in the local data are unlabeled. Therefore, we utilize only the labeled nodes in the local training set to train the synthesizer. Meanwhile, to ensure that the synthesized graph faithfully reflects both the distribution of node features and intrinsic topological patterns of the original graph, we design a two-fold consistency mechanism to compute the synthesizing objective $\mathcal{L}_{\text{syn}}$. Details can be found in Appendix B .

**Topological Codebook.** While the Synergy Graph Synthesizer effectively aligns global feature distributions and preserves large-scale structural patterns, it inevitably smooths out nuanced distinctions among fine-grained substructures. To address this limitation, we introduce a discrete structural latent space—termed the *Topological Codebook*—to explicitly model subtle neighborhood variations. First, each node $v_i \in \mathcal{V}$ is mapped to a continuous embedding $\mathbf{h}_i \in \mathbb{R}^D$ via an encoder. We then construct a learnable codebook $\mathbf{E}_{\varphi''} = [\mathbf{e}_1, \ldots, \mathbf{e}_M] \in \mathbb{R}^{M \times D}$, where each code vector $\mathbf{e}_j$ represents a distinct local substructure pattern. For each node $v_i$, the discrete token corresponding to its local substructure is determined by quantizing its continuous embedding to the nearest code:

$$z_i = \arg\min_{j \in \{1, \ldots, M\}} \|\mathbf{h}_i - \mathbf{e}_j\|_2^2. \tag{6}$$

This quantization process maps continuous embeddings into discrete tokens $\{z_1, \ldots, z_N\}$, where each node is assigned a token corresponding to its most similar local substructure. Using these tokens, we derive the quantized representations $\mathbf{Q} = \{\mathbf{e}_{z_1}, \cdots, \mathbf{e}_{z_N}\}$ for all nodes. These discrete representations enable the model to capture subtle variations in node neighborhoods more effectively.

**Concurrent Optimization.** To learn the Topological Codebook, we adapt the core principles of Vector Quantized-Variational Autoencoders (VQ-VAE) for graph data. The model optimizes the

following components: the reconstruction loss $\mathcal{L}_{\text{syn}}$ and two additional losses: the consistency loss and the discretization loss. The consistency loss ensures that the selected code $\mathbf{e}_{z_i}$ preserves original node attributes and topology information. Specifically, we design two decoders $p_\gamma, p_{\gamma'}$ to separately map the quantized representations to the original dimension: $\mathbf{e}'_{z_i} = p_\gamma(\mathbf{e}_{z_i}), \mathbf{Q}' = p_{\gamma'}(\mathbf{Q})$. Then, we introduce our consistency loss:

$$\mathcal{L}_{\text{con}} = \frac{1}{N} \sum_{i=1}^{N} ||\mathbf{e}'_{z_i} - x_i||_2^2 + ||\mathbf{A} - \sigma(\mathbf{Q}' \cdot \mathbf{Q}'^{\top}))||_2^2, \tag{7}$$

where $\sigma$ denotes the sigmoid function. Meanwhile, the discretization loss consists of three key components: the codebook term, the commitment term, and the orthogonal term. The codebook term ensures that the selected code $\mathbf{e}_{z_i}$ is aligned with the encoder output $\mathbf{h}_i$, maintaining the coherence between continuous and discrete representations. The commitment term encourages the encoder output to stay close to the chosen code, preventing excessive fluctuations between code vectors that could destabilize the learning process. By stabilizing the encoding process, the commitment term ensures that each node's representation remains consistent with its assigned code. The orthogonal term promotes diversity among codebook vectors by encouraging them to be independent and avoiding convergence in the same direction:

$$\mathcal{L}_{\text{disc}} = \frac{1}{N} \sum_{i=1}^{N} \left\| \text{sg}[\mathbf{h}_i] - \mathbf{e}_{z_i} \right\|_2^2 + \frac{\eta}{N} \sum_{i=1}^{N} \left\| \text{sg}[\mathbf{e}_{z_i}] - \mathbf{h}_i \right\|_2^2 + \lambda_o \left( \frac{1}{M^2} \sum_{i,j}^{M} \left( \frac{\mathbf{e}_i^{\top} \mathbf{e}_j}{||\mathbf{e}_i|| ||\mathbf{e}_j||} \right)^2 - \frac{1}{M} \right), \tag{8}$$

where $\text{sg}[\cdot]$ denotes the stop-gradient operator, and $\eta, \lambda_o$ are hyperparameters controlling the strength of each term. The overall optimization objective combines these losses with a negative log-likelihood term $\mathcal{L}_{\text{NLL}}\left(y_i, \hat{y}_{\mathbf{e}_{z_i}}\right)$ to maintain node-label consistency with GNN $\mathcal{F}_\theta$ predictions $\hat{y}_{\mathbf{e}_{z_i}}$:

$$\mathcal{L}_{\text{overall}} = \mathcal{L}_{\text{syn}} + \mathcal{L}_{\text{NLL}}(y_i, \hat{y}_{\mathbf{e}_{z_i}}) + \mathcal{L}_{\text{con}} + \lambda_c \mathcal{L}_{\text{disc}}, \tag{9}$$

where $\lambda_c$ is a balancing hyperparameter. By jointly optimizing these terms, we obtain a Topological Codebook $\mathbf{E}_{\varphi''}$ that captures a rich set of discrete tokens, each representing a unique local substructure. This latent structural space empowers the model to integrate localized topological knowledge with global graph structures. The concurrent optimization of the Synthesizer $\mathcal{Q}_{\varphi'}$ and the Topological Codebook leads to the creation of a refined knowledge repository $\mathcal{R}_\varphi^\star = \{\mathcal{Q}_{\varphi'}^\star, \mathbf{E}_{\varphi''}^\star\}$, which effectively synthesizes both micro- and macro-level graph information. In the subsequent section, we explore how this extracted knowledge can be transferred to construct a generalizable global model.

### 3.3 Server Graph Knowledge Integration

**Wasserstein-Enhanced Semantic Affinity Distillation.** After local training, each client uploads its respective knowledge base $\mathcal{R}_\varphi^k = \{\mathcal{Q}_{\varphi'}^k, \mathbf{E}_{\varphi''}^k\}$ and local GNN $\mathcal{F}_\theta^k$ to the central server. At the server, for each local model $k$, the local GNN model $\mathcal{F}_\theta^k$ serves as the teacher, while the global GNN model $\mathcal{F}_\phi$ acts as the student. Specifically, a proxy graph $\hat{\mathcal{G}}^k$ is generated via $\mathcal{Q}_{\varphi'}^k(\epsilon; \hat{y}_{\text{uni}})$, where $\hat{y}_{\text{uni}}$ denotes a class-balanced distribution. This proxy graph retains general information from the client data distribution, thereby providing an effective signal for subsequent knowledge transfer.

A central challenge in this knowledge transfer process is minimizing the loss incurred between the teacher and student models. Traditional distillation techniques based on Kullback-Leibler divergence perform only category-to-category comparisons, failing to capture nuanced cross-category semantic affinities. To address this limitation, we propose the *Wasserstein-Enhanced Semantic Affinity Distillation (WESAD)* method, which employs the discrete Wasserstein distance to achieve a comprehensive alignment between the probability distributions of the teacher and student models. For the $k$-th teacher GNN model, let $p_T^k$ denote the class probability distribution produced by $\mathcal{F}_\theta^k$ and $p_S$ the distribution produced by $\mathcal{F}_\phi$. These distributions are computed via the softmax function $\sigma$ with temperature $\tau$. To measure their discrepancy, we define the discrete Wasserstein distance loss $D_{\text{WESAD}}^k$ as follows:

$$D_{\text{WESAD}}^k(p_T^k, p_S) = \min_{q_{ab}} \sum_{a,b} c_{ab}^k q_{ab} + \eta q_{ab} \log q_{ab}, \tag{10}$$

where $q_{ab}$ represents the mass transferred from the teacher's category $C_a$ to the student's category $C_b$, subject to the constraints:

$$q_{ab} \geq 0, \quad \sum_b q_{ab} = p_{T,a}^k, \quad \sum_a q_{ab} = p_{S,b}^k. \tag{11}$$

Here, $\eta$ is a hyperparameter controlling the entropy regularization term. A key component of this formulation is the cost matrix $c_{ab}^k$, which encapsulates the semantic dissimilarity between categories. We convert the semantic affinity, denoted as $\mathrm{SA}^k(C_a, C_b)$, into a distance metric by defining

$$\mathrm{SA}^k(C_a, C_b) = W^k[C_a][:] \otimes W^k[C_b][:], \tag{12}$$

where $\otimes$ denotes element-wise multiplication, and $W^k$ is the weight matrix of the teacher's projection head after $l_2$ normalization. The term $\mathrm{SA}^k(C_a, C_b)$ reflects the intrinsic semantic affinity between categories in the local GNN teacher $\mathcal{F}_{\theta^k}$. We then compute the cost matrix $c_{ab}^k$ as

$$c_{ab}^k = 1 - \kappa \mathrm{SA}^k(C_a, C_b), \tag{13}$$

where $\kappa$ controls the cost. When two categories are semantically similar, a higher $\mathrm{SA}^k(C_a, C_b)$ results in a lower $c_{ab}^k$, thereby reducing the transportation cost between them; conversely, semantically dissimilar categories incur a higher cost. This design ensures that probability mass is effectively reallocated between semantically proximate categories during knowledge transfer, facilitating fine-grained semantic alignment. By minimizing $D_{\mathrm{WESAD}}^k(p_T^k, p_S)$, the global student model $\mathcal{F}_\phi$ is not only aligned with the output distribution of each local teacher model $\mathcal{F}_\theta^k$ but also benefits from the incorporation of inter-class semantic correlations.

**Wasserstein-Driven Structural Relation Distillation.** Building upon the semantic knowledge transfer described before, we now seek to integrate the fine-grained latent structural knowledge. Unlike semantic alignment, which primarily focuses on matching class-level distributions, structural distillation emphasizes the nuanced topological patterns underlying node interactions. We introduce a *Wasserstein-Driven Structural Relation Distillation (WDSRD)* scheme that harmonizes the structural articulations by leveraging the topological codebook $\mathbf{E}_{\varphi''}^k = [\mathbf{e}_1^k, \ldots, \mathbf{e}_M^k]$, where each $\mathbf{e}_m^k \in \mathbb{R}^D$ is a learnable structural token that represents a prototypical substructure. This codebook unifies latent structural information for both local and global models. Concretely, for each generated proxy graph $\hat{\mathcal{G}}^k$, we compute two sets of node-level representations:

$$\hat{\mathbf{H}}_{\mathrm{local}}^k = \mathcal{F}_\theta^k(\hat{\mathcal{G}}^k) \in \mathbb{R}^{\hat{N}^k \times D}, \quad \hat{\mathbf{H}}_{\mathrm{global}} = \mathcal{F}_\phi(\hat{\mathcal{G}}^k) \in \mathbb{R}^{\hat{N}^k \times D}, \tag{14}$$

where $\hat{N}^k$ denotes the number of nodes in $\hat{\mathcal{G}}^k$ and $D$ is the feature dimension. For each node $v_i$, let $h_i^k \in \mathbb{R}^D$ be its representation from the local model and $h_i \in \mathbb{R}^D$ from the global model. We then compare each $\hat{h}_i^k$ (or $\hat{h}_i$) against all $M$ codes in $\mathbf{E}_{\varphi''}^k$ to derive a soft assignment distribution:

$$\mathcal{B}_i^k = \sigma(\mathrm{Dist}(\hat{h}_i^k, \mathbf{E}_{\varphi''}^k)/\tau), \quad \widetilde{\mathcal{B}}_i^k = \sigma(\mathrm{Dist}(\hat{h}_i, \mathbf{E}_{\varphi''}^k)/\tau), \tag{15}$$

where $\mathrm{Dist} : \mathbb{R}^D \times \mathbb{R}^{M \times D} \longrightarrow \mathbb{R}^M$ is a distance-based comparison function (*e.g.*, Euclidean distance) that assigns each node $v_i$ to every code $\mathbf{e}_m$ with a probability reflecting their similarity. We then obtain structural code assignments $\widetilde{\mathcal{B}}_i^k$ from global GNN and $\mathcal{B}_i^k$ from local GNN over the $M$ structural codes from $k$-th topological codebook $\mathbf{E}_{\varphi''}^k$. We first define the mean as the simple average of the assignment vectors across all nodes. Specifically, the local and global means are given by:

$$\mu_{\mathrm{local}}^k = \frac{1}{\hat{N}^k} \sum_{i=1}^{\hat{N}^k} \mathcal{B}_i^k, \quad \mu_{\mathrm{global}}^k = \frac{1}{\hat{N}^k} \sum_{i=1}^{\hat{N}^k} \widetilde{\mathcal{B}}_i^k. \tag{16}$$

To characterize the dispersion of the assignments, we compute the covariance matrices as:

$$\Sigma_{\mathrm{local}}^k = \frac{1}{\hat{N}^k} \sum_{i=1}^{\hat{N}^k} \left(\mathcal{B}_i^k - \mu_{\mathrm{local}}^k\right) \left(\mathcal{B}_i^k - \mu_{\mathrm{local}}^k\right)^\top,$$
$$\Sigma_{\mathrm{global}}^k = \frac{1}{\hat{N}^k} \sum_{i=1}^{\hat{N}^k} \left(\widetilde{\mathcal{B}}_i^k - \mu_{\mathrm{global}}^k\right) \left(\widetilde{\mathcal{B}}_i^k - \mu_{\mathrm{global}}^k\right)^\top. \tag{17}$$

Thus, the local and global assignments for the agent graph are approximated by the Gaussian distributions: $\mathcal{N}\left(\mu_{\mathrm{local}}^k, \Sigma_{\mathrm{local}}^k\right), \mathcal{N}\left(\mu_{\mathrm{global}}^k, \Sigma_{\mathrm{global}}^k\right)$. Inspired by the Wasserstein distance, we employ the closed-form Wasserstein distance between two Gaussian distributions. For two Gaussians, $\mathcal{N}(\mu_1, \Sigma_1)$ and $\mathcal{N}(\mu_2, \Sigma_2)$, the Wasserstein distance is decomposed into a term measuring the difference in means and a term capturing the difference in covariances:

$$D_{\mathrm{WD}}\left(\mathcal{N}(\mu_1, \Sigma_1), \mathcal{N}(\mu_2, \Sigma_2)\right) = \|\mu_1 - \mu_2\|_2$$
$$+ \mathrm{tr}\left(\Sigma_1 + \Sigma_2 - 2\left(\Sigma_1^{1/2}\Sigma_2\Sigma_1^{1/2}\right)^{1/2}\right). \tag{18}$$

Table 1: **Comparison with the state-of-the-art methods** on eight real-world datasets. We report node classification accuracies (%) for downstream task performance. Green arrows $_\uparrow$ denote advancements in accuracy metrics than FedAvg while red arrows $_\downarrow$ indicate regressions. OOM means out-of-memory error. The best and second results are highlighted with **bold** and underline, respectively.

| Category | Methods | Cora | CiteSeer | PubMed | Amz-Photo | Coauthor-CS | Actor | Roman-Empire | Obgn-Arxiv |
|---|---|---|---|---|---|---|---|---|---|
| FL | FedAvg [ASTAT17] | 30.61 | 32.88 | 57.91 | 23.12 | 22.50 | 24.40 | 18.49 | 14.58 |
| | FedProx [MLSys20] | $30.98_{\uparrow0.37}$ | $\underline{35.73}_{\uparrow2.85}$ | $50.56_{\downarrow7.35}$ | $24.16_{\uparrow1.04}$ | $21.44_{\downarrow1.06}$ | $23.75_{\downarrow0.65}$ | $15.46_{\downarrow3.03}$ | $13.99_{\downarrow0.59}$ |
| | FedNova [NeurIPS20] | $14.21_{\downarrow16.40}$ | $18.58_{\downarrow17.30}$ | $33.48_{\downarrow24.43}$ | $6.15_{\downarrow16.97}$ | $18.83_{\downarrow3.67}$ | $20.04_{\downarrow4.36}$ | $6.49_{\downarrow12.00}$ | $1.17_{\downarrow13.41}$ |
| | FedRCL [CVPR24] | $17.60_{\downarrow13.01}$ | $12.73_{\downarrow20.15}$ | $28.12_{\downarrow29.79}$ | $4.92_{\downarrow18.20}$ | $14.75_{\downarrow7.75}$ | $10.72_{\downarrow13.68}$ | $11.47_{\downarrow7.02}$ | $2.56_{\downarrow12.02}$ |
| FGL | FedPub [ICML23] | $30.52_{\downarrow0.09}$ | $34.91_{\uparrow2.03}$ | $41.22_{\downarrow16.69}$ | $21.91_{\downarrow1.21}$ | $\underline{26.75}_{\uparrow4.25}$ | $22.17_{\downarrow2.23}$ | $13.71_{\downarrow4.78}$ | $10.02_{\downarrow4.56}$ |
| | FGSSL [IJCAI23] | $30.23_{\downarrow0.38}$ | $21.95_{\downarrow10.93}$ | $39.68_{\downarrow18.23}$ | $13.06_{\downarrow10.06}$ | $22.44_{\downarrow0.06}$ | $22.19_{\downarrow2.21}$ | $14.62_{\downarrow1.87}$ | $9.24_{\downarrow5.34}$ |
| | FedGTA [VLDB24] | $14.02_{\downarrow16.59}$ | $17.75_{\downarrow15.13}$ | $31.45_{\downarrow26.46}$ | $4.10_{\downarrow19.02}$ | $10.80_{\downarrow11.70}$ | $19.25_{\downarrow5.15}$ | $4.55_{\downarrow13.94}$ | $1.15_{\downarrow13.43}$ |
| | FedTAD [IJCAI24] | $30.43_{\downarrow0.18}$ | $33.86_{\uparrow0.98}$ | $39.32_{\downarrow18.59}$ | $22.01_{\downarrow1.11}$ | $14.09_{\downarrow8.41}$ | $23.58_{\downarrow0.82}$ | $14.40_{\downarrow4.09}$ | OOM |
| OFL | DENSE [NeurIPS22] | $12.92_{\downarrow17.71}$ | $7.87_{\downarrow25.01}$ | $20.84_{\downarrow37.07}$ | $4.93_{\downarrow18.19}$ | $3.96_{\downarrow18.54}$ | $10.72_{\downarrow13.68}$ | $3.90_{\downarrow14.59}$ | $0.33_{\downarrow14.25}$ |
| | FedCVAE [ICLR23] | $30.89_{\uparrow0.28}$ | $34.76_{\uparrow1.88}$ | $52.01_{\downarrow5.90}$ | $\underline{31.62}_{\uparrow8.50}$ | $14.60_{\downarrow7.90}$ | $19.38_{\downarrow5.02}$ | $\underline{28.99}_{\uparrow10.50}$ | $13.71_{\downarrow0.87}$ |
| | FedSD2C [NeruIPS24] | $17.78_{\downarrow12.83}$ | $29.96_{\downarrow2.92}$ | $26.12_{\downarrow31.79}$ | $8.73_{\downarrow14.39}$ | $3.88_{\downarrow18.62}$ | $18.83_{\downarrow5.57}$ | $3.92_{\downarrow14.57}$ | $0.76_{\downarrow13.82}$ |
| | FENS [NeruIPS24] | $\underline{31.43}_{\uparrow0.82}$ | $20.97_{\downarrow11.91}$ | $49.07_{\downarrow8.84}$ | $25.30_{\uparrow2.18}$ | $22.54_{\downarrow0.04}$ | $24.30_{\downarrow0.10}$ | $13.96_{\downarrow4.53}$ | $13.09_{\downarrow1.49}$ |
| OFGL | OASIS | $\mathbf{49.59}_{\uparrow18.98}$ | $\mathbf{45.69}_{\uparrow12.81}$ | $\mathbf{60.99}_{\uparrow3.08}$ | $\mathbf{63.73}_{\uparrow40.61}$ | $\mathbf{60.44}_{\uparrow37.94}$ | $\mathbf{25.42}_{\uparrow1.02}$ | $\mathbf{30.07}_{\uparrow11.58}$ | $\mathbf{15.05}_{\uparrow0.47}$ |

Accordingly, the $D_{\text{WDSRD}}$ between the local and global assignments for the entire agent graph is

$$
\begin{aligned}
D_{\text{WDSRD}}^k = & \left\| \mu_{\text{local}}^k - \mu_{\text{global}}^k \right\|_2 \\
& + \text{tr}\left( \Sigma_{\text{local}}^k + \Sigma_{\text{global}}^k - 2\left( \left(\Sigma_{\text{local}}^k\right)^{\frac{1}{2}} \Sigma_{\text{global}}^k \left(\Sigma_{\text{local}}^k\right)^{\frac{1}{2}} \right)^{\frac{1}{2}} \right),
\end{aligned}
\tag{19}
$$

where the first term quantifies the discrepancy between the local and global means, while the second term reflects the differences in the distribution shapes as captured by the covariances. Minimizing $\mathcal{L}_{\text{WDSRD}}$ encourages the local and global GNNs to progressively align their output distributions of structural assignments, thereby achieving an effective distillation of fine-grained topological knowledge.

In our final server optimization, we integrate both the distillation losses into a single objective with addition of NLL loss on the proxy graph:

$$
L_{\text{server}} = \sum_{k=1}^{K} \frac{\hat{N}^k}{\hat{N}} \left( \mathcal{L}_{\text{NLL}}^k + \lambda_k D_{\text{WESAD}}^k + \lambda_s \tau^2 D_{\text{WDSRD}}^k \right),
\tag{20}
$$

where $\hat{N}$ denotes the overall size of all synthesized graphs and $\lambda_k, \lambda_s$ balance each term. This unified approach leverages the strengths of both methods: WESAD captures and aligns inter-class semantic affinities to transfer rich semantic signals, while WDSRD preserves the fine-grained topological structures inherent in local graphs. Minimizing this combined loss ensures that the global model assimilates both semantic and structural knowledge from the local teachers, resulting in a more generalizable global model.

## 4 Experiment

In this section, we comprehensively evaluate OASIS through four axes: **Q1** (Superiority), **Q2** (Effectiveness), **Q3** (Sensitivity), **Q4** (Pricacy). The answers of **Q1-Q2** are illustrated in Sec. 4.2-Sec. 4.3 and the analyses of **Q3-Q4** can be found in Appendix K and Appendix L.

### 4.1 Experimental Setup

**Datasets.** To effectively evaluate the performance of our approach, we employed eight benchmark graph datasets of various scales and distributions, including Cora [37], CiteSeer [10], PubMed [5], Amazon-Photo, Coauthor-CS [45], Actor [42], Roman-empire [43] and Ogbn-Arxiv [17]. Detailed descriptions and splits for these datasets can be found in Appendix D.

**Counterparts.** We compare OASIS against several traditional FL methods: (1) **FedAvg** [ASTAT17] [38], (2) **FedProx** [MLSys20] [27], (3) **FedNova** [NeurIPS20] [54], (4) **FedRCL** [CVPR24] [44]; four popular FGL approaches: (5) **FedPub** [ICML23] [4], (6) **FGSSL** [IJCAI23] [18]; (7) **FedTAD** [IJCAI24] [73], (8) **FedGTA** [VLDB24] [30]; four One-shot FL methods: (9) **DENSE** [NeurIPS22] [66], (10) **FedCVAE** [ICLR23] [15], (11) **FedSD2C** [NeurIPS24] [67], (12) **FENS** [NeurIPS24] [1]. Detailed descriptions can be found in Appendix E.

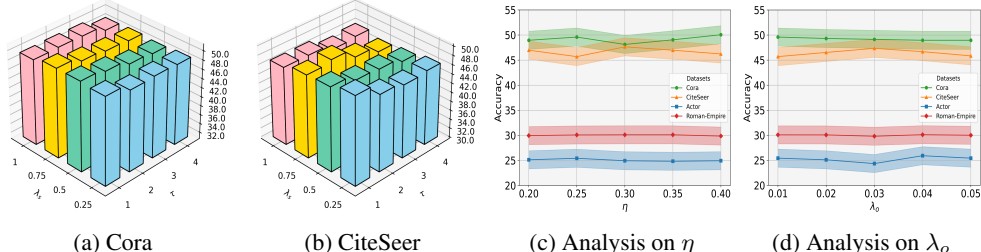

| (a) Cora | (b) CiteSeer | (c) Analysis on $\eta$ | (d) Analysis on $\lambda_o$ |

Figure 4: **Sensitivity Study** of hyperparameters. In (a) and (b), we vary $\tau$ and $\lambda_s$ on Cora and CiteSeer respectively. In (c) and (d), we conduct sensitive study of $\eta$ and $\lambda_o$ on four datasets. Please refer to Appendix K for further analysis.

**Implementation Details.** We adopt a two-layer GCN as the backbone, with a hidden layer size of 128. We set $K = 10$ clients and draw $p_k \sim \text{Dir}(\alpha)$ from a Dirichlet distribution [40] and assign a fraction $p_k^c$ of class $c$ to client $k$. Specifically, $\alpha$ is set as 0.05 to simulate a highly non-IID senario. The codebook size is set in the range $\{2^6, 2^7, 2^8\}$. More implementation details and experiments on various client numbers can be found in Appendix F and Appendix G.

## 4.2 Superiority

To address **Q1**, we analyze the superior performance of `OASIS`. We demonstrate the node classification performance with various real-world graph datasets and summarize the generalized test accuracy in Tab. 1. From the table, several key observations can be made (**Obs.**): **Obs. ❶** `OASIS` consistently outperforms other counterparts, with an average improvement of 15.81% over the baseline. By capturing fine-grained structural information through our synthesizer and codebook, the global model is able to acquire more intricate knowledge during distillation. **Obs. ❷** Traditional FL and FGL methods heavily depend on gradual local model updates over multiple rounds, and some are tailored for personalized optimization, such as FedPub and FedGTA. Therefore, clients fail to efficiently share generalized information within one communication round, leading to the poor model performance. **Obs. ❸** One-shot FL approaches such as FedCVAE and FedSD2C perform well in some small graphs. However, they excessively rely on class-related semantic signals and overlook the structural information, thus struggling with large-scale graphs.

## 4.3 Effectiveness

To address **Q2**, we conduct an ablation study on the key components in the Server Graph Knowledge Integration part: *Semantic Affinity Distillation* (WESAD) and the *Topological Codebook* (WESRD). The experimental results on Cora and Amazon-Photo are shown in Figure 5. From the bar chart, we can observe that both components improve the model performance significantly. WESAD employs the Wasserstein distance to align the probability distribution between the teacher (local GNN) and the student (global GNN), highlighting the incorporation of inter-class semantic correlations. Meanwhile,

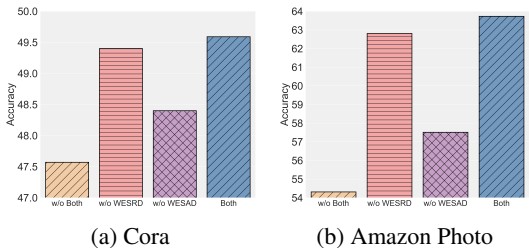

| (a) Cora | (b) Amazon Photo |

Figure 5: **Ablation Study** of the Semantic Affinity Distillation (WESAD) and Topological Codebook (WESRD) on Cora and Amazon-Photo datasets. For an in-depth analysis, please refer to Sec. 4.3.

WDSRD integrates the fine-grained structural knowledge from the topological codebook. When both WESAD and WDSRD are combined, the performance reaches its peak, with both semantic and structural knowledge effectively distilled to the well-generalizable student global model.

## 5 Conclusion

In this paper, we introduce `OASIS` to address two key challenges in existing One-shot Federated Learning approaches with generative models: weak awareness of fine-grained structural knowledge and poor distillation capability from the topological aspect. We first establish a Synergy Graph Synthesizer to capture complex structural knowledge and then construct a structural latent space by

introducing the Topological Codebook. On the server side, we propose Wasserstein-Enhanced Semantic Affinity Distillation to model inter-class relationships and build Wasserstein-Driven Structural Relation Distillation to precisely transfer intricate topological knowledge from the codebook to the global model. Extensive experiments on diverse datasets demonstrate the effectiveness of `OASIS`.

## Acknowledgement

This work is supported by National Natural Science Foundation of China under Grant (62361166629, 62225113, 623B2080), the Major Project of Science and Technology Innovation of Hubei Province (2024BCA003, 2025BEA002), and the Innovative Research Group Project of Hubei Province under Grants 2024AFA017. The supercomputing system at the Supercomputing Center of Wuhan University supported the numerical calculations in this paper.

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

# A Notations

We present a comprehensive review of the commonly used notations and their definitions in Tab. 2.

Table 2: Notation and Definitions

| Notation | Definition |
| --- | --- |
| $\mathcal{G}$ | Graph data. |
| $\mathcal{V}$ | The node set of $\mathcal{G}$. |
| $\mathcal{E}$ | The edge set of $\mathcal{G}$. |
| $\mathbf{X}$ | The feature matrix of $\mathcal{G}$. |
| $\mathbf{A}$ | The adjacency matrix of $\mathcal{G}$. |
| $F$ | The dimension of the node feature. |
| $D$ | The dimension of hidden embeddings and codebook tokens. |
| $K$ | The number of clients. |
| $\hat{\mathcal{G}}^k$ | The generated graph for client $k$. |
| $\hat{\mathbf{X}}$ | The synthetic feature matrix. |
| $\mathbf{H}$ | The activated similarity matrix. |
| $\hat{\mathbf{A}}$ | The synthetic adjacency matrix. |
| $v_i$ | Node $i$ in $\mathcal{V}$. |
| $h_i^l$ | The representation of $v_i$ at the $l$-th layer of GNN. |
| $\mathcal{N}(v_i)$ | The set of neighbours of node $v_i$. |
| $\mathcal{F}_{\theta^k}$ | The local model of client $k$. |
| $\theta^k$ | The parameters of the local model $\mathcal{F}_{\theta^k}$ of client $k$. |
| $\mathcal{F}_\phi$ | The global model. |
| $\mathcal{F}_\phi^\star$ | The ultimate global model. |
| $\phi$ | The parameters of the global model. |
| $\mathcal{R}_\varphi^\star$ | The optimal knowledge repository. |
| $\mathcal{R}_\varphi^k$ | The specialized knowledge repository of client $k$. |
| $\epsilon$ | The Gaussian noise. |
| $\mathbf{E}_{\varphi''}^k$ | The topological codebook of client $k$. |
| $M$ | The number of tokens in the codebook. |
| $z_i$ | The discrete token index in the codebook. |
| $\mathbf{Q}$ | The quantized representations. |
| $\mathcal{Q}_{\varphi'}^k$ | The Synergy Graph Synthesizer of client $k$. |
| $\hat{y}_{\text{uni}}$ | The class-balanced distribution. |
| $p_T^k$ | The class probability distribution produced by $\mathcal{F}_{\theta^k}$. |
| $p_\mathcal{S}$ | The class probability distribution produced by $\mathcal{F}_\phi$. |
| $\tau$ | The temperature. |
| $C_a$ | The teacher's category. |
| $C_b$ | The student's category. |
| $c_{ab}^k$ | The $k$-th cost matrix. |
| $q_{ab}$ | The mass transferred from $C_a$ to $C_b$. |
| $W^k$ | The weight matrix of the projection head of client $k$'s teacher GNN after $l_2$ normalization. |
| $\mathcal{B}_i^k$ | The structural code assignments from the teacher GNN. |
| $\tilde{\mathcal{B}}_i^k$ | The structural code assignments from the global GNN. |
| $\hat{N}^k$ | The size of the synthesized graph of client $k$. |
| $\hat{N}$ | The overall size of all synthesized graphs. |
| $\mu_{\text{local}}^k$ | The local means of the assignment $\mathcal{B}_i^k$. |
| $\mu_{\text{global}}^k$ | The global means of the assignment $\tilde{\mathcal{B}}_i^k$. |
| $\Sigma_{\text{local}}^k$ | The covariance matrix of $\mathcal{B}_i^k$. |
| $\Sigma_{\text{global}}^k$ | The covariance matrix of $\tilde{\mathcal{B}}_i^k$. |

# B Alignment of the Synergy Graph Synthesizer

To ensure that the synthesized graph faithfully reflects both the statistical distribution of node features and the intrinsic topological patterns of the original graph, we design a two-fold consistency mechanism. First, to align the distributions of synthetic and original features, we introduce a distribution convergence term:

$$\mathcal{L}_{\text{dist}} = \frac{1}{|\mathcal{V}|} \sum_{i=1}^{|\mathcal{V}|} \sum_{f=1}^{F} s_i(f) \log \frac{s_i(f)}{\hat{s}_i(f)}, \tag{21}$$

where $s_i = \text{softmax}(x_i)$ and $\hat{s}_i = \text{softmax}(\hat{x}_i)$ represent the normalized representations of the original and synthetic features, respectively. Second, to enhance local feature consistency and mitigate mode collapse, we incorporate a feature dispersion constraint:

$$\mathcal{L}_{\text{spread}} = 1 - \frac{1}{|\mathcal{V}|} \sum_{i=1}^{|\mathcal{V}|} \frac{\hat{x}_i \cdot x_i}{\|\hat{x}_i\|_2 \cdot \|x_i\|_2}. \tag{22}$$

Moreover, to simultaneously reconcile structural and feature discrepancies, we employ a fused Gromov-Wasserstein loss that quantifies the differences between the original and synthetic graphs within an optimal transport framework:

$$\mathcal{L}_{\text{FGW}} = \min_{\mathbf{\Gamma}(\mu,\hat{\mu})} \sum_{i,j,u,v} \Big( a\big(\mathbf{A}(i,j) - \hat{\mathbf{A}}(u,v)\big)^2$$
$$+ (1-a)\|\mathbf{X}(i) - \hat{\mathbf{X}}(u)\|_2^2 \Big) \mathbf{\Gamma}_{i,u} \mathbf{\Gamma}_{j,v}, \tag{23}$$

where $a \in [0,1]$ regulates the relative importance of structural and feature fidelity, and the transport plan $\mathbf{\Gamma}$ is determined under the assumption of uniform marginal distributions $\mu$ and $\hat{\mu}$. These components are integrated into a unified synthesizing objective:

$$\mathcal{L}_{\text{syn}} = \mathcal{L}_{\text{spread}} + \lambda_d \mathcal{L}_{\text{dist}} + \lambda_f \mathcal{L}_{\text{FGW}}, \tag{24}$$

where hyperparameters $\lambda_d$ and $\lambda_f$ control the relative contributions of each term. By leveraging the Synergy Graph Synthesizer, each client generates a refined synthetic graph. However, this approach relies on a continuous latent space and employs techniques such as KNN-based adjacency construction, which primarily capture global statistical trends and ensure overall structural consistency. Consequently, these methods tend to aggregate partial connectivity patterns, inadvertently smoothing fine-grained details and diminishing subtle local nuances.

# C Related Work

**Federated Graph Learning.** Federated Graph Learning (FGL) extends Federated Learning (FL) to graph-structured data, enabling decentralized training while preventing the exposure of raw graph data, thus enhancing privacy protection [14? ? ]. Existing FGL methods can be categorized into intra-graph FGL and inter-graph FGL [65]. Inter-graph FGL approaches such as GCFL+ [59] and FedGNN [56] consider settings where clients possess disjoint graphs, such as molecular graphs or independent social networks, and focus on training separate graph models for each client. In contrast, intra-graph FGL assumes that each client holds a subgraph of a globally connected graph, with methods such as FedSSP [50] and FGGP [52] aiming to aggregate local updates while maintaining connectivity. However, these approaches struggle to capture fine-grained structural variations and cross-client dependencies. Moreover, their dependence on iterative communication leads to high costs. To address these issues, we propose a one-shot FGL approach that removes iterative communication while preserving fine-grained structural knowledge through a novel hierarchical knowledge distillation framework, thereby improving model generalization.

**One-shot Federated Learning.** One-shot Federated Learning (OFL) is a paradigm that reduces communication overhead by limiting the number of communication rounds to one, making it particularly advantageous for resource-constrained and privacy-sensitive environments [12, 34]. Unlike traditional federated learning methods that require multiple iterative updates [38, 28], OFL eliminates prolonged client-server interactions, significantly reducing latency and computational overhead. Several recent methods seek to enhance OFL through generative modeling techniques, such as FedDEO [61] and Dense [66], or ensemble-based strategies, such as FuseFL [51] and FENS [1]. Additionally, knowledge distillation techniques have been introduced to facilitate more effective cross-client knowledge

transfer in OFL, as seen in FedDF [33] and FedGEMS [6]. However, existing OFL methods fail to effectively capture local semantics and fine-grained structural variations in federated graph learning. Consequently, we introduce a Synergy Graph Synthesizer to align global features while preserving large-scale structures and a Topological Codebook to model neighborhood variations, improving graph representation in one-shot FGL.

**Knowledge Distillation.** Knowledge Distillation (KD) is widely utilized for model compression and knowledge transfer [16], allowing smaller models to perform comparably to larger models while reducing computational demands. In federated learning, methods such as FD-FAug [21, 60] extend KD techniques to address non-independent and identically distributed (non-IID) data. Similarly, KD has been leveraged in OFL, preserving knowledge transfer, where clients distill local knowledge into a compact representation for global aggregation. Existing OFL distillation methods can be classified into data-based and model-based distillation [34]. Data-based approaches, including DOSFL [71], FedD3 [48], and FedSD2C [67], utilize synthetic data to transfer knowledge across clients. Conversely, model-based distillation methods transfer knowledge through latent feature compressions, such as Dense [66] and FedCVAE [15]. However, existing methods fail to preserve intricate graph structures, while traditional KD techniques weaken inter-class semantic affinities, limiting their effectiveness in heterogeneous FL. Thus, this study introduces a Hierarchical Wasserstein-based Distillation framework that aligns semantic affinities and preserves structural dependencies, facilitating efficient and privacy-preserving knowledge transfer.

## D  Dataset Details.

To assess the effectiveness of , we conduct experiments on eight real-world graph datasets: Cora, CiteSeer, PubMed, Amazon-Photo, CoAuthor-CS, Actor, Roman-Empire, and Ogbn-Arxiv. Each dataset is split into training, validation, and test sets in a fixed 20%/40%/40% ratio. The key statistics of these datasets are summarized in Tab. 3. A detailed description is provided below:

- **Cora, CiteSeer, and PubMed.** These three citation network datasets are standard benchmarks in graph-based machine learning, especially for tasks like node classification and link prediction. In these datasets, nodes correspond to academic papers, while edges represent citation links. Each node is assigned a class label, and its feature vector is constructed from textual information such as words in the title or abstract. These datasets exhibit sparsity and high dimensionality, making them well-suited for evaluating the effectiveness and scalability of graph neural networks (GNNs).
- **Amazon-Photo.** This dataset is built from the Amazon product catalog, where nodes represent product images and edges indicate co-purchase relationships. Each photo is categorized into a specific class, and node features are derived from image metadata. Amazon-Photo serves as a benchmark for testing graph-based learning models in visual domains.
- **CoAuthor-CS.** This dataset represents a co-authorship network in the field of computer science, where nodes correspond to research papers, and edges denote co-authorship relations. Each paper is associated with a topic category, and features are extracted from the paper's title and abstract. This dataset is commonly used to evaluate node classification and community detection algorithms.
- **Actor.** The Actor dataset is a heterophilic graph where nodes represent actors, and edges indicate their co-occurrence on the same Wikipedia page. Node features are derived from textual descriptions, and classification is performed based on predefined actor categories. This dataset presents unique challenges due to its structural differences from traditional citation networks.
- **Roman-empire.** The Roman-empire dataset captures historical relationships in an ancient setting, where nodes correspond to different entities, and edges represent interactions between them. The dataset is particularly useful for studying graph-based algorithms in non-traditional network structures, offering a distinct perspective on real-world graph learning.
- **Ogbn-Arxiv.** This large-scale citation network is constructed from arXiv papers, where nodes represent papers and edges capture citation links. Each paper belongs to a specific subject category, including physics, computer science, and mathematics. Node features are extracted from paper abstracts. Ogbn-Arxiv is widely used for benchmarking GNNs due to its diversity and scale.

## E  Counterpart Details.

This section provides a comprehensive overview of the baseline approaches employed in our study.

Table 3: **Statistics** of datasets used in experiments.

| Dataset | #Nodes | #Edges | #Classes | #Features |
|---|---|---|---|---|
| Cora | 2,708 | 5,278 | 7 | 1,433 |
| Citeseer | 3,327 | 4,552 | 6 | 3,703 |
| Pubmed | 19,717 | 44,324 | 3 | 500 |
| Amz-Photo | 7,650 | 287,326 | 8 | 745 |
| Coauthor-CS | 18,333 | 327,576 | 15 | 6,805 |
| Actor | 7600 | 30,019 | 5 | 932 |
| Roman-empire | 22,622 | 65,854 | 18 | 300 |
| Obgn-Arxiv | 169,343 | 1,166,243 | 40 | 128 |

- **FedAvg** [ASTAT17]. A foundational algorithm in Federated Learning, FedAvg operates by allowing clients to independently train models on their local datasets and subsequently transmit their model updates to a central server. The server performs a weighted aggregation of these updates to refine the global model, which is then redistributed to the clients for further local training. By transmitting only model parameters instead of raw data, FedAvg reduces communication costs and enhances privacy. However, it struggles with performance degradation in scenarios where client data distributions are highly non-IID [29, 39].
- **FedProx** [MLSys20]. As an enhancement of FedAvg, FedProx is specifically designed to address the challenges posed by statistical heterogeneity in federated learning. It introduces an additional regularization term that constrains local updates, preventing excessive divergence from the global model. This proximal term mitigates the impact of local data distribution shifts, leading to more stable convergence. By ensuring consistency in updates across clients, FedProx demonstrates improved robustness in non-IID settings.
- **FedNova** [NeurIPS20]. FedNova refines the FedAvg framework by introducing normalization to local updates before aggregation. Unlike standard averaging methods, FedNova ensures that each client's contribution to the global model is proportional to the amount of data it possesses. This approach addresses the issue of unequal client influence, leading to more balanced and efficient convergence. FedNova is particularly beneficial in federated environments where data distributions are skewed across clients.
- **FedRCL** [CVPR24]. FedRCL incorporates contrastive learning to improve federated learning performance under data heterogeneity [63, 53]. It examines inconsistencies in gradient updates across clients and attributes them to variations in feature distributions. To counteract this, FedRCL employs a contrastive regularization strategy that penalizes overly similar samples within a class, ensuring diverse and transferable feature representations. This approach enhances collaborative learning among clients and leads to notable performance improvements.
- **FedPub** [ICML23]. Unlike traditional FL methods that focus on training a single global model, Fed-Pub adopts a personalized approach by facilitating the interaction of local Graph Neural Networks (GNNs). It employs functional embeddings to quantify similarity between client models, enabling an adaptive weighted aggregation at the server. Furthermore, a sparse mask mechanism allows clients to selectively update subgraph-relevant parameters, improving both privacy preservation and learning efficiency in heterogeneous graph scenarios.
- **FGSSL** [IJCAI23]. FGSSL addresses local client distortion caused by both node-level semantics and graph-level structures. It improves discrimination by contrasting nodes from different classes, aligning local nodes with their global counterparts of the same class while pushing them away from different classes. To handle structural information, it transforms adjacency relationships into similarity distributions and distills relational knowledge from the global model into local models. This approach preserves both structural integrity and discriminability, achieving superior performance on multiple graph datasets.
- **FedTAD** [IJCAI24]. Designed to address subgraph heterogeneity in federated learning, FedTAD decomposes variations in local graphs into differences in label distributions and structural homophily. By analyzing these discrepancies, it prevents misleading model aggregation, which can occur when local models contribute inconsistently. The framework enhances knowledge transfer through topology-aware knowledge distillation, improving both reliability and aggregation efficiency in FL settings.
- **FedGTA** [VLDB24]. FedGTA is tailored for large-scale graph federated learning, tackling issues of slow convergence and suboptimal scalability. Unlike prior methods that focus on either optimization

strategies or complex local models, FedGTA integrates topology-aware local smoothing with mixed neighbor feature aggregation to improve learning efficiency [72]. By leveraging graph structures in aggregation, it enhances scalability and performance in federated graph learning.

- **DENSE** [NeurIPS22]. A framework designed to overcome limitations of conventional one-shot FL, DENSE eliminates the need for additional auxiliary datasets or model information by employing a two-stage learning process. It first synthesizes data representations and then applies model distillation to refine the global model. This approach ensures that a federated model can be effectively trained in a single round of communication while accommodating heterogeneous client architectures.

- **FedCVAE** [ICLR23]. A data-free one-shot FL method, FedCVAE utilizes a Conditional Variational Autoencoder (CVAE) [8, 24, 23] to improve generalization under statistical heterogeneity. The approach reframes local learning objectives, allowing effective global aggregation despite distribution disparities. An extended variant, FedCVAE-KD, incorporates knowledge distillation [11] to consolidate local decoders into a unified global model. FedCVAE outperforms traditional baselines, particularly in highly heterogeneous settings.

- **FedSD2C** [NeurIPS24]. FedSD2C is a novel one-shot FL framework that mitigates performance degradation caused by data heterogeneity. It leverages a distillation-based strategy to synthesize informative data representations directly from local distributions, bypassing the inconsistency issues present in conventional model aggregation. By sharing distilled representations instead of raw model updates, FedSD2C enhances knowledge transfer and ensures greater consistency across federated clients.

- **FENS** [NeurIPS24]. FENS is a novel approach to OFL that aims to bridge the accuracy gap between standard federated learning (FL) and OFL while maintaining high communication efficiency. FENS employs a two-phase learning process: first, clients train local models and send them to the server, as in OFL; second, clients collaboratively train a lightweight prediction aggregator using FL. Extensive experiments demonstrate that FENS achieves performance close to FL while preserving the efficiency of OFL.

## F   Implementation Details.

The experiments are conducted using NVIDIA GeForce RTX 3090 GPUs as the hardware platform, coupled with Intel(R) Xeon(R) Gold 6240 CPU @ 2.60GHz. The deep learning framework employed was Pytorch, version 2.3.1, alongside CUDA version 12.1. We adopt a two-layer GCN as the backbone, with a hidden layer size of 128. Moreover, we utilize 3 hidden linear layers and a projection head as the synergic graph synthesizer which concats random noise and one-hot label as the input and generates pseudo features as the output. We set $K = 10$ clients and draw $p_k \sim \mathrm{Dir}(\alpha)$ from a Dirichlet distribution [40] and assign a fraction $p_k^c$ of class $c$ to client $k$. As for optimization of the graph synthesizer, Adaptive Moment Estimation (Adam) was chosen, featuring a learning rate of $5e-3$ and a weight decay of $4e-4$. The codebook size is set in the range $\{2^6, 2^7, 2^8\}$, with the same optimizer and learning parameter. At the local training phase, we set the training epoch $T_S$ of the synthesizer to 100 and epoch $T_C$ of the teacher GNN and the codebook to 50. $\lambda_d$ and $\lambda_f$ are determined through a grid search [32] within $\{0.01, 0.05, 0.1, 0.5\}$ and $\{0.1, 0.2, 0.5, 1\}$ respectively. $\eta, \lambda_o$ are set as $0.25, 0.01$ and $\lambda_c$ is set to 1. To make sure that $\mathcal{L}_{\mathrm{FGW}}$ is on the same scale as other loss functions for Amz-Photo and Ogbn-Arxiv datasets, we set their $\lambda_f$ scales to $1e-5$ and $1e-7$, respectively. We set $a$ in $\mathcal{L}_{\mathrm{FGW}}$ as $0.5$ to balance the feature part and the structure part. The communication round is limited to **one**. At the server side, we determine the global distilling epoch in the range $\{10, 20, 30\}$ and adopt Adam as the optimizer for the global model with a learning rate of $1e-2$ and a weight decay of $4e-4$. The synthesized graphs generated by the synthesizer of each client have the same scale $\hat{N}^k$ as the corresponding local subgraph. For the large graph Ogbn-Arxiv, we set $T_C$ to 1 and $\hat{N}^k$ to one-tenth of the local graph. Moreover, $\lambda_k, \kappa$ is set to $0.01, 1$ and $\lambda_s$ is determined in range $\{1, 5, 10\}$. The distillation temperature $\tau$ is set to 3 for all datasets.

## G   Ablation Study on Different Numbers of Clients.

In this section, we vary the number of clients in $\{5, 10, 20\}$ and conduct the node classification task on CiteSeer, Amazon-Photo. Experimental Results are shown in Tab. 4. From the table, we can observe that our `OASIS` outperforms all counterparts with different numbers of clients, demonstrating the stability of `OASIS` across various data distributions and subgraph scales. As we simulate a highly

Table 4: **Comparison with the state-of-the-art methods with different numbers of clients.** We report node classification accuracies (%) for downstream task performance. Green arrows ↑ denote advancements in accuracy metrics than FedAvg while red arrows ↓ indicate regressions. OOM means out-of-memory error. The best and second results are highlighted with **bold** and underline, respectively.

| Datasets (→) | | CiteSeer | | | Amazon-Photo | | |
|---|---|---|---|---|---|---|---|
| Category | Methods (↓) | 5 Clients | 10 Clients | 20 Clients | 5 Clients | 10 Clients | 20 Clients |
| FL | FedAvg [ASTAT17] | 35.43 | 32.88 | 38.13 | 49.93 | 23.12 | 21.79 |
| | FedProx [MLSys20] | $40.32_{\uparrow 4.89}$ | $\underline{35.73}_{\uparrow 2.85}$ | $\underline{39.39}_{\uparrow 1.26}$ | $46.54_{\downarrow 3.39}$ | $24.16_{\uparrow 1.04}$ | $23.58_{\uparrow 1.79}$ |
| | FedNova [NeurIPS20] | $18.43_{\downarrow 17.00}$ | $18.58_{\downarrow 14.30}$ | $16.17_{\downarrow 21.96}$ | $9.69_{\downarrow 40.24}$ | $6.15_{\downarrow 16.97}$ | $8.57_{\downarrow 13.22}$ |
| | FedRCL [CVPR24] | $15.88_{\downarrow 19.55}$ | $12.73_{\downarrow 20.15}$ | $7.05_{\downarrow 31.08}$ | $21.80_{\downarrow 28.13}$ | $4.92_{\downarrow 18.20}$ | $10.69_{\downarrow 11.10}$ |
| FGL | FedPub [ICML23] | $20.07_{\downarrow 15.36}$ | $34.91_{\uparrow 2.03}$ | $32.27_{\downarrow 5.86}$ | $42.00_{\downarrow 7.93}$ | $21.91_{\downarrow 1.21}$ | $21.76_{\downarrow 0.03}$ |
| | FGSSL [IJCAI23] | $20.82_{\downarrow 14.61}$ | $21.95_{\downarrow 10.93}$ | $34.79_{\downarrow 3.34}$ | $41.64_{\downarrow 8.29}$ | $13.06_{\downarrow 10.06}$ | $22.28_{\uparrow 0.49}$ |
| | FedGTA [VLDB24] | $16.93_{\downarrow 18.50}$ | $17.75_{\downarrow 15.13}$ | $15.73_{\downarrow 22.40}$ | $5.39_{\downarrow 44.54}$ | $4.10_{\downarrow 19.02}$ | $6.17_{\downarrow 15.62}$ |
| | FedTAD [IJCAI24] | $19.25_{\downarrow 16.18}$ | $33.86_{\uparrow 0.98}$ | $28.34_{\downarrow 9.79}$ | $25.29_{\downarrow 24.64}$ | $22.01_{\downarrow 1.11}$ | $21.57_{\downarrow 0.22}$ |
| OFL | DENSE [NeurIPS22] | $7.64_{\downarrow 27.79}$ | $7.87_{\downarrow 25.01}$ | $7.06_{\downarrow 31.07}$ | $4.96_{\downarrow 44.97}$ | $4.93_{\downarrow 18.19}$ | $4.90_{\downarrow 16.89}$ |
| | FedCVAE [ICLR23] | $\underline{40.33}_{\uparrow 4.90}$ | $34.76_{\uparrow 1.88}$ | $18.10_{\downarrow 20.03}$ | $\underline{52.39}_{\downarrow 2.54}$ | $\underline{31.62}_{\uparrow 8.50}$ | $24.87_{\uparrow 3.08}$ |
| | FedSD2C [NeruIPS24] | $20.97_{\downarrow 14.46}$ | $29.96_{\downarrow 2.92}$ | $23.66_{\downarrow 14.47}$ | $11.75_{\downarrow 38.18}$ | $8.73_{\downarrow 14.39}$ | $23.38_{\uparrow 1.59}$ |
| | FENS [NeruIPS24] | $21.50_{\downarrow 13.93}$ | $20.97_{\downarrow 11.91}$ | $17.95_{\downarrow 20.18}$ | $22.26_{\downarrow 27.67}$ | $25.30_{\uparrow 2.18}$ | $\underline{25.53}_{\uparrow 3.74}$ |
| OFGL | `OASIS` | $\mathbf{40.45}_{\uparrow 5.02}$ | $\mathbf{45.69}_{\uparrow 12.81}$ | $\mathbf{47.92}_{\uparrow 9.79}$ | $\mathbf{71.18}_{\uparrow 21.25}$ | $\mathbf{63.73}_{\uparrow 40.61}$ | $\mathbf{54.11}_{\uparrow 32.32}$ |

non-IID scenario using a Dirichlet distribution with a small concentration parameter $\alpha = 0.05$, increasing the number of clients may lead to situations where some clients possess very limited data or even no training samples at all, which does not fully align with real-world settings. Nevertheless, as shown in Tab. 4, `OASIS` consistently maintains a stable and effective performance trend, even under the presence of a larger number of clients.

# H   Performance of OASIS under moderate or mild heterogeneity

To demonstrate the performance of `OASIS` under moderate or mild heterogeneity (or even i.i.d. data), we have conducted additional experiments with varying levels of data heterogeneity. Specifically, we evaluate the performance of `OASIS` under different values of the Dirichlet distribution parameter $\alpha \in \{1, 10, 100, 1000, 10000\}$ to simulate a range of heterogeneity scenarios. These experiments are performed on the Cora dataset, and we adopt FedAvg, FedPub, FedCVAE and FedGCN(1-hop & 2-hop) for comparison. Notably, when $\alpha$=10000, the data distribution approximates an i.i.d. setting.

Table 5: Performance of `OASIS` under milder heterogeneity (or even i.i.d.) scenarios.

| $\alpha$ | 1 | 10 | 100 | 1000 | 10000 |
|---|---|---|---|---|---|
| FedAvg | 49.50 | 56.23 | 57.93 | 57.84 | 45.11 |
| FedPub | 39.87 | 44.96 | 60.13 | 52.70 | 58.55 |
| FedCVAE | 60.45 | 56.50 | 59.58 | 55.27 | 56.63 |
| FedGCN (1-hop) | 52.06 | 41.39 | 50.60 | 51.97 | 56.91 |
| FedGCN (2-hop) | 58.66 | 52.11 | 55.82 | 57.56 | 61.76 |
| **OASIS** | **63.15** | **75.64** | **76.35** | **69.11** | **68.44** |

As shown Tab. 5, `OASIS` consistently outperforms the existing methods across various heterogeneity levels, including strong, moderate, mild, and even i.i.d. conditions. The advantage of `OASIS` remains robust even as the heterogeneity decreases, demonstrating that its performance is not diminished by a shift towards more homogeneous data. We will ensure that this additional experimental result is included in the revision.

The reason we choose to focus on strong heterogeneity in our experiments is that, in real-world scenarios, non-uniform data distribution is more prevalent and presents a greater challenge. We aim to showcase that `OASIS` not only performs well in typical i.i.d. and mild heterogeneity settings but also excels in more challenging scenarios with high heterogeneity.

# I Performance of OASIS on other prominent GNN models

To demonstrate the performance of `OASIS` on other GNN models, we conducted experiments on the Cora and CiteSeer datasets using GAT and GraphSAGE respectively. We compare `OASIS` against five baselines. Results are shown in Tab. 6.

Table 6: The performance of OASIS on GAT and GraphSAGE.

| Methods | Cora-GAT | Cora-GraphSAGE | CiteSeer-GAT | CiteSeer-GraphSAGE |
|---------|----------|----------------|--------------|--------------------|
| FedAvg  | 30.98    | 29.88          | 30.49        | 37.23              |
| FedProx | 31.71    | 29.97          | 27.94        | 39.40              |
| FGSSL   | 29.79    | 27.45          | 20.52        | 20.22              |
| FedCVAE | 29.61    | 25.94          | 22.05        | 23.52              |
| FENS    | 29.70    | 29.88          | 21.05        | 21.50              |
| **OASIS** | **42.62** | **39.96**   | **38.73**    | **44.87**          |

From the table, we can observe that our `OASIS` consistently outperforms other baselines in both GAT and GraphSAGE backbones, demonstrating the capability of `OASIS` to generalize to other prominent GNN models.

# J Mathematical Analysis of OASIS

Here we provide a thorough analysis on the mathematical bound on information retention of the synthesizer and the impact of global distillation.

## J.1 Mathematical Bound on Information Retention

Let $\mathcal{G}_l = (\mathcal{V}_l, \mathcal{E}_l, X_l)$ be the local graph, where $\mathcal{V}_l$ is the node set, $\mathcal{E}_l$ the edge set, and $X_l \in \mathbb{R}^{|\mathcal{V}_l| \times d}$ the node feature matrix. The synthesized graph is $\mathcal{G}_s = (\mathcal{V}_s, \mathcal{E}_s, X_s)$, with $X_s \in \mathbb{R}^{|\mathcal{V}_s| \times d}$ and adjacency matrix $A_s$. The goal is to ensure that $\mathcal{G}_s$ preserves the critical information in $\mathcal{G}_l$, including node feature distributions and topological structures.

To quantify the retention of critical information, we derive a bound on the divergence between the local graph $\mathcal{G}_l \sim \mathcal{D}_l$ and the synthesized graph $\mathcal{G}_s \sim \mathcal{D}_s$, using the Synthesizer's loss as a proxy. We measure the divergence between their distributions using a combined metric that accounts for both feature and topological differences.

**Step 1: Feature Distribution Divergence.** The KL divergence loss $L_{\text{dist}} = D_{\text{KL}}(P_{X_s} \| P_{X_l})$ directly measures the feature distribution mismatch. By Pinsker's inequality, the total variation distance is bounded by:

$$\delta_{\text{TV}}(P_{X_s}, P_{X_l}) \leq \sqrt{\frac{1}{2} D_{\text{KL}}(P_{X_s} \| P_{X_l})} = \sqrt{\frac{1}{2} L_{\text{dist}}}.$$

Assuming $L_{\text{dist}} \leq \epsilon_d$, the feature distributions are close in total variation:

$$\delta_{\text{TV}}(P_{X_s}, P_{X_l}) \leq \sqrt{\frac{\epsilon_d}{2}}.$$

**Step 2: Feature and Topological Alignment via FGW.** The Fused Gromov-Wasserstein loss $L_{\text{fgw}}$ bounds the combined feature and topological mismatch. The FGW distance can be decomposed as:

$$L_{\text{fgw}} = \text{FGW}(\mathcal{G}_l, \mathcal{G}_s) = \alpha W_2^2(P_{X_l}, P_{X_s}) + (1 - \alpha)\text{GW}(A_l, A_s),$$

where $W_2^2$ is the squared 2-Wasserstein distance between feature distributions, and GW is the Gromov-Wasserstein distance between structures. Assuming $L_{\text{fgw}} \leq \epsilon_f$, we have:

$$\alpha W_2^2(P_{X_l}, P_{X_s}) + (1 - \alpha)\text{GW}(A_l, A_s) \leq \epsilon_f.$$

Since both terms are non-negative, we get:

$$W_2(P_{X_l}, P_{X_s}) \le \sqrt{\frac{\epsilon_f}{\alpha}}, \quad \mathrm{GW}(A_l, A_s) \le \frac{\epsilon_f}{1 - \alpha}.$$

The Wasserstein distance further bounds the feature distribution divergence, and the Gromov-Wasserstein distance bounds the structural discrepancy.

**Step 3: Feature Diversity.** The cosine similarity loss $L_{\text{spread}}$ ensures that the synthesized features do not collapse. We assume $L_{\text{spread}} \ge -\eta$, where $\eta > 0$ is a constant reflecting sufficient feature spread. This prevents degenerate solutions where $X_s^i \approx X_s^j$ for all $i, j$.

**Step 4: Combined Bound.** To derive a bound on the overall divergence between $\mathcal{D}_l$ and $\mathcal{D}_s$, we consider a joint metric that combines feature and topological differences. Define a graph divergence metric:

$$\Delta(\mathcal{G}_l, \mathcal{G}_s) = W_2(P_{X_l}, P_{X_s}) + \mathrm{GW}(A_l, A_s).$$

From the FGW loss:

$$\Delta(\mathcal{G}_l, \mathcal{G}_s) \le W_2(P_{X_l}, P_{X_s}) + \mathrm{GW}(A_l, A_s) \le \sqrt{\frac{\epsilon_f}{\alpha}} + \frac{\epsilon_f}{1 - \alpha} = \epsilon_f \left( \sqrt{\frac{1}{\alpha}} + \frac{1}{1 - \alpha} \right).$$

The KL divergence provides an additional constraint on features. Combining with Pinsker's inequality, the total variation distance on features is:

$$\delta_{\mathrm{TV}}(P_{X_s}, P_{X_l}) \le \sqrt{\frac{\epsilon_d}{2}}.$$

Since Wasserstein and total variation distances are related (e.g., via transport inequalities in bounded spaces), we focus on the FGW-based bound for simplicity, as it captures both features and topology.

**Theorem 1.** Assuming the Synthesizer's loss is bounded as $L_{\text{syn}} \le \epsilon$, with $L_{\text{dist}} \le \epsilon_d$, $L_{\text{fgw}} \le \epsilon_f$, and $L_{\text{spread}} \ge -\eta$, the divergence between the local and synthesized graph distributions is bounded as:

$$\Delta(\mathcal{D}_l, \mathcal{D}_s) \le \epsilon_f \left( \sqrt{\frac{1}{\alpha}} + \frac{1}{1 - \alpha} \right) + \sqrt{\frac{\epsilon_d}{2}},$$

where $\Delta(\mathcal{D}_l, \mathcal{D}_s) = \mathbb{E}_{\mathcal{G}_l \sim \mathcal{D}_l, \mathcal{G}_s \sim \mathcal{D}_s}[W_2(P_{X_l}, P_{X_s}) + \mathrm{GW}(A_l, A_s)]$.

**Proof.** The total synthesis loss is:

$$L_{\text{syn}} = L_{\text{spread}} + \lambda_d L_{\text{dist}} + \lambda_f L_{\text{fgw}} \le \epsilon.$$

Since $L_{\text{spread}} \ge -\eta$, we have:

$$\lambda_d L_{\text{dist}} + \lambda_f L_{\text{fgw}} \le \epsilon + \eta.$$

Assume $L_{\text{dist}} \le \epsilon_d$, $L_{\text{fgw}} \le \epsilon_f$, with $\lambda_d \epsilon_d + \lambda_f \epsilon_f \le \epsilon + \eta$. The feature divergence is bounded via Pinsker's inequality:

$$\delta_{\mathrm{TV}}(P_{X_s}, P_{X_l}) \le \sqrt{\frac{\epsilon_d}{2}}.$$

The FGW loss bounds the combined feature and topological divergence:

$$\alpha W_2^2(P_{X_l}, P_{X_s}) + (1 - \alpha)\mathrm{GW}(A_l, A_s) \le \epsilon_f.$$

Thus:

$$W_2(P_{X_l}, P_{X_s}) \le \sqrt{\frac{\epsilon_f}{\alpha}}, \quad \mathrm{GW}(A_l, A_s) \le \frac{\epsilon_f}{1-\alpha}.$$

The total divergence is:

$$\Delta(\mathcal{D}_l, \mathcal{D}_s) \le \mathbb{E}\left[W_2(P_{X_l}, P_{X_s}) + \mathrm{GW}(A_l, A_s)\right] \le \sqrt{\frac{\epsilon_f}{\alpha}} + \frac{\epsilon_f}{1-\alpha}.$$

Incorporating the feature distribution bound from $L_{\mathrm{dist}}$, we add the total variation term for completeness, yielding:

$$\Delta(\mathcal{D}_l, \mathcal{D}_s) \le \epsilon_f\left(\sqrt{\frac{1}{\alpha}} + \frac{1}{1-\alpha}\right) + \sqrt{\frac{\epsilon_d}{2}}.$$

This bound quantifies the retention of critical information: a small $\epsilon_f$ and $\epsilon_d$ ensure that the synthesized graph's feature and topological distributions are close to those of the local graph.

The divergence between the local and synthesized graph distributions is bounded by:

$$\Delta(\mathcal{D}_l, \mathcal{D}_s) \le \epsilon_f\left(\sqrt{\frac{1}{\alpha}} + \frac{1}{1-\alpha}\right) + \sqrt{\frac{\epsilon_d}{2}},$$

where $\Delta(\mathcal{D}_l, \mathcal{D}_s)$ combines feature and topological differences. This bound rigorously quantifies the retention of critical information, ensuring that the synthesized graph effectively captures the local graph's semantic and structural properties when $\epsilon_d$ and $\epsilon_f$ are small.

## J.2 Impact on Global Model Generalization

The retention of critical information and the distillation mechanisms directly influence the generalization performance of the global model. Our `OASIS` employs two distillation mechanisms to transfer local knowledge to the global model:

- **WESAD**: Transfers inter-class semantic relationships.
- **WDSRD**: Transfers fine-grained topological structures.

**Generalization Bound.** To quantify the impact on generalization, we derive a bound on the global model's expected error using the Rademacher complexity framework, adapted for federated graph learning.

Let $\mathcal{D}_l$ denote the local data distribution for client $l$, and $\mathcal{D}_s$ the distribution of the synthesized data. The global model $f_g \in \mathcal{F}$ (a hypothesis class of graph neural networks) is trained on synthesized data to minimize the empirical risk:

$$\hat{R}_s(f_g) = \mathbb{E}_{\mathcal{D}_s}[\ell(f_g(\mathcal{G}_s), y)],$$

where $\ell$ is the loss function (e.g., cross-entropy), and $y$ is the label. The true risk is:

$$R(f_g) = \mathbb{E}_{\mathcal{D}_l}[\ell(f_g(\mathcal{G}_l), y)].$$

The generalization gap is $R(f_g) - \hat{R}_s(f_g)$. Assuming the synthesized graph retains critical information (i.e., $W_1(\mathcal{D}_l, \mathcal{D}_s) \le \epsilon$), we bound the generalization error using the Wasserstein distance and Rademacher complexity.

**Theorem 2.** For a hypothesis class $\mathcal{F}$ with Rademacher complexity $\mathcal{R}_n(\mathcal{F})$ over $n$ samples, and assuming the loss function $\ell$ is $L$-Lipschitz, the expected generalization error of the global model is bounded as:

$$\mathbb{E}[R(f_g)] \le \hat{R}_s(f_g) + 2L\mathcal{R}_n(\mathcal{F}) + L\epsilon + C\sqrt{\frac{\log(1/\delta)}{n}},$$

with probability at least $1 - \delta$, where $\epsilon = W_1(\mathcal{D}_l, \mathcal{D}_s)$ is the Wasserstein distance between local and synthesized distributions, and $C$ is a constant.

The term $L\epsilon$ quantifies the impact of information retention: if $\epsilon$ is small (i.e., the synthesized graph closely matches the local graph), the generalization error is tightly bounded.

**Proof.** By the Wasserstein distance property, the difference in expected loss is bounded:

$$|\mathbb{E}_{\mathcal{D}_l}[\ell(f_g(\mathcal{G}_l), y)] - \mathbb{E}_{\mathcal{D}_s}[\ell(f_g(\mathcal{G}_s), y)]| \le LW_1(\mathcal{D}_l, \mathcal{D}_s) = L\epsilon.$$

Using standard generalization bounds for empirical risk minimization:

$$\mathbb{E}[R(f_g)] \le \hat{R}_s(f_g) + 2\mathcal{R}_n(\mathcal{F}) + C\sqrt{\frac{\log(1/\delta)}{n}}.$$

Combining these, we obtain:

$$\mathbb{E}[R(f_g)] \le \hat{R}_s(f_g) + 2L\mathcal{R}_n(\mathcal{F}) + L\epsilon + C\sqrt{\frac{\log(1/\delta)}{n}}.$$

# K  Sensitivity

To address **Q3**, we conduct analyses on hyperparameters of `OASIS`. Specifically, we compare the model performance under different values of temperature $\tau$ and $\lambda_c$. We vary $\tau$ and $\lambda_c$ in range $[1, 4]$ and $[0.25, 1]$ with 1 and 0.25 as the step size respectively. Moreover, as for the Topology Codebook, we vary $\eta$ and $\lambda_o$ in range $[0.2, 0.4]$ and $[0.01, 0.05]$ with 0.05 and 0.01 as the step size. Results shown in Figure 4 demonstrate that the performance of `OASIS` stays consolidated under different hyperparameter values, proving the robustness of `OASIS`.

# L  Privacy Security

## L.1  How to prevent malicious attackers from stealing communication data?

Privacy security plays a crucial role in FGL systems. In Sec. 3.2, we propose a novel Synergy Graph Synthesizer $\mathcal{Q}_{\varphi'}$ to generate powerful synthesized graphs with labels and Gaussian noise $\epsilon$. For simplicity, a standard normal distribution is typically utilized, where the center $\Omega$ is set to 0. However, we introduce a shift $\varsigma$ to the center $\Omega$ at the client side and communicate $\varsigma$ to the server either offline or through encryption methods [15, 70]. We consider the worst-case scenario where an eavesdropping attacker intercepts all the parameters. However, without knowledge of the specific shift $\varsigma$, the attacker can only utilize the original center $\hat{\Omega}$, which is distant from the true center $(\Omega + \varsigma)$. Alternatively, the attacker might attempt to overlap the center using a wide uniform distribution.

To simulate these situations, we first set the original noise distribution center $\Omega = 0$ with shift $\varsigma$ varies in $\{30, 60, 90\}$ while the attacker still takes $\hat{\Omega} = 0$. We conduct all experiments here on the node classification task with the knowledge distillation part excluded to explicitly demonstrate the influence of the generated data. Results are shown in Figure 6a. From the bar chart, we can observe that the model performance sharply declined by $60\%$, proving that the data generated with center $\hat{\Omega}$ are totally different from local data. Moreover, to simulate the overlapping attempt, we set $\Omega + \varsigma = 0$ and the overlap range in $\{\mathcal{U}(-100, 100), \mathcal{U}(-200, 200), \mathcal{U}(-300, 300)\}$ and conduct the experiment in the same setting. Results are shown in Figure 6b, with the same phenomenon observed. Therefore, we prove the security of data privacy of our `OASIS`.

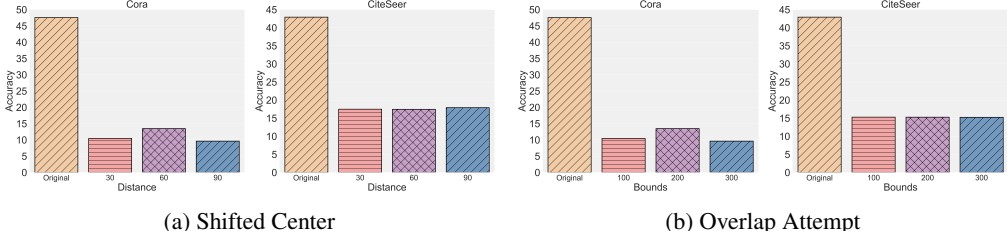

(a) Shifted Center          (b) Overlap Attempt

Figure 6: **Privacy Study** of `OASIS`. **Original** here means the server generates the graphs with the same distribution center as clients. For an in-depth analysis, please refer to Appendix L.1.

### L.2 How to prevent a curious server from inspecting client privacy?

Although we design an encrypted shift $\varsigma$ for the distribution center $\Omega$, a curious server can still utilize the correct distribution to synthesize data. However, the true local data information remains protected and is not exposed to the server, due to the following reasons:

**(1) Discrepancy in Label Distributions.** In practical FGL scenarios, the label distribution across clients is typically highly non-identical. In our experiments, we simulate an extremely non-IID setting by sampling labels from a Dirichlet distribution with concentration parameter $\alpha = 0.05$. Importantly, clients do not transmit their actual label distribution to the server. Instead, the data synthesis is conducted under the assumption of a uniform label distribution $\hat{y}_{\text{uni}}$ (Sec. 3.3). Consequently, the overall label distribution of the synthesized graph on the server deviates significantly from that of any individual client, thereby mitigating the risk of direct data leakage.

**(2) Structural Dissimilarity of Nodes.** Even for synthesized nodes that share the same label as some local data, their structural context (e.g., neighborhood connectivity) is distinct. As described in Equation (5), each node in the synthesized graph is connected to its five most similar nodes based on feature similarity. This fixed K-nearest-neighbor (KNN) construction introduces structural differences compared to the true graph topology of local data. Although our alignment module (Appendix B) encourages the preservation of structural semantics via $\mathcal{L}_{\text{FGW}}$, it does not enforce strict local topological isomorphism. Therefore, structural privacy is preserved to a considerable degree.

**(3) Feature Perturbation via Two-Fold Consistency Mechanism.** The synthesizer is trained with a two-fold consistency objective $\mathcal{L}_{\text{syn}}$ involving both feature and structure alignments (Equation (24)). This objective steers the optimization away from directly replicating raw node features. As a result, even nodes with the same labels as in local data will exhibit distinct feature representations in the synthesized graph.

In summary, privacy preservation in our framework is inherently balanced with the learning objectives of the Synergy Graph Synthesizer. Through the structural regularization from the KNN construction and the two-fold consistency mechanism, we ensure that the synthesized data avoids leaking sensitive client information, while still capturing high-level latent knowledge required for effective global model learning. This design satisfies the fundamental goals of federated graph learning.

## M Discussion on Limitations.

While `OASIS` achieves notable success in efficiently capturing fine-grained structural knowledge of local graphs and effectively transfer the knowledge during server distillation in the one-shot scenario, it still has inherent limitations as a sythesizer-based method [49]. In particular, the presence of noise in local data [22] can impair the ability of the synthesizer to effectively extract and learn local-specific patterns, which may in turn impede the distillation module. Improving the robustness of the Synergy Graph Synthesizer against such noise interference [35, 26] remains a promising direction for future work.

## N Discussion on Broader Impacts

Our proposed `OASIS` framework contributes to the broader field of FGL by enabling more efficient and privacy-preserving collaboration across decentralized graph datasets, especially under stringent communication constraints. By introducing domain-specific generative strategies and novel structural distillation techniques, `OASIS` opens new possibilities for applying OFL to graph-structured data such

as social networks, biomedical graphs, and knowledge graphs, where data are inherently sensitive and dispersed. This advancement can benefit applications involving privacy-critical domains like healthcare, finance, and cybersecurity, empowering institutions to jointly learn high-quality models without exposing private graph data. As FL technologies become more widely adopted, continued attention to the ethical implications of synthetic data generation and the interpretability of structural knowledge transfer will be essential to fostering responsible AI deployment.

