# OpenReview forum: "OASIS: One-Shot Federated Graph Learning via Wasserstein Assisted Knowledge Integration"
_NeurIPS.cc/2025/Conference — NeurIPS 2025 poster_

### Official Review · Reviewer_4jz9 · 2025-06-23

**Clarity:** 2
**Significance:** 2
**Originality:** 2
**Rating:** 4
**Confidence:** 3

**Summary:**

The paper introduces OASIS, a novel one-shot Federated Graph Learning (FGL) framework designed to address the challenges of capturing fine-grained structural knowledge from local graphs and effectively transferring this knowledge to the global model during knowledge distillation. By integrating the Synergy Graph Synthesizer, Topological Codebook, and Wasserstein-driven distillation techniques, the framework aims to enhance both the performance and generalization ability of federated graph models in resource-constrained settings. The authors present a comprehensive set of experiments, showing that OASIS outperforms existing methods in terms of node classification accuracy across a variety of real-world datasets.

**Questions:**

1. Why the accuracy is lower than other works?

2. Can OASIS be compared to fedsage+?

3. Can you provide a more detailed analysis of the communication cost compared to methods like fedGCN? or fedsage+?

4. Can you added formal theoretical justification or guarantees on the possible information losses?

**Ethical Concerns:**

["NO or VERY MINOR ethics concerns only"]

**Final Justification:**

Most of my concerns are resolved. However, the theoretical analysis is still limited on strict assumptions and the assumptions lack realistic justification. The analysis in response is like mechanically stacking blocks together, lacking deeper insight.
Although I dont unserstand why reviewer BhNM appricate this mathematic description and say that  it facilitates theoretical understanding even thougn this paper dose not have any theoretically analysis before, I agree that this paper has superior experimental preformance over existing methods.
Since I am not expert in FGL, I am not sure whether a FGL paper without theoretical justification can be accepct or not. Therefore, I will slightly raise my score.

**Limitations:**

yes

**Quality:**

2

**Strengths And Weaknesses:**

Strengths:
1.The paper presents a well-structured framework, OASIS, which integrates novel techniques such as the Synergy Graph Synthesizer and Wasserstein-based knowledge distillation to address fine-grained structural knowledge transfer in federated learning. This is a significant contribution to the field of One-shot Federated Graph Learning (OFL).

2.The paper conducts a comprehensive set of experiments on several real-world graph datasets, including Cora, CiteSeer, and PubMed, demonstrating the potential of OASIS in improving global model performance. However, discrepancies between the results reported here and in other works may need further investigation.

Weaknesses:
1. A key concern is the substantial drop in accuracy compared to the original works. For example, the accuracy reported for FedPub on the Cora dataset is only 30%, compared to 79% in the original paper. This discrepancy requires careful analysis and clarification.

2. Can OASIS be considered a data enhancement method? It should be compared with FedSAGE+ [1], an elegant method that also trains the model on generated graphs. A thorough analysis of the differences between these approaches, or an explanation of why OASIS is superior is preferred.

3. Although the paper presents OASIS as a one-shot federated learning framework, the results do not convincingly demonstrate how the communication overhead is reduced in practical settings. The need to upload both the model parameters and the Topological Codebook could lead to higher communication costs compared to other methods, such as FedGCN [2], which communicate with the central server in a single pre-training step. A more detailed analysis of communication efficiency is needed to assess the practical scalability of the method.

4.Despite OASIS incorporating multiple components (e.g., Synergy Graph Synthesizer, Topological Codebook, and WDSRD), the paper does not provide sufficient theoretical justification for why these blocks improve performance. A theoretical analysis of the difference between the OASIS-generated graph and the original graph is needed to better bound the performance of OASIS. This could either provide insights into how much OASIS outperforms the original method or offer a theoretical understanding of the potential performance loss compared to directly transferring the locally trained model on the original clean graph.

[1] https://arxiv.org/pdf/2106.13430

[2] https://arxiv.org/pdf/2201.12433

---

> ### Author Rebuttal · Authors · 2025-07-29
>
> ***Dear Reviewer 4jz9:***
>
> We appreciate your time and thoughtful feedback. We have addressed your concerns below and hope that the clarifications will positively contribute to the evaluation of our work.
>
> **`Weakness 1` & `Question 1`: Why the accuracy is lower compared to the original works (e.g. FedPub) ?**
>
> Thank you for  your insightful question about the experimental results. In Section 4.2, we have already discussed the poor performance of some baselines, and we would like to provide further clarification here.
>
> First, it is important to highlight that our experiments were conducted under extreme data heterogeneity conditions. Specifically, we employed a Dirichlet distribution with $\alpha = 0.05$, which leads to highly skewed data distributions across clients. This level of heterogeneity makes it extremely difficult for FL and FGL methods to effectively capture local knowledge during training. Moreover, they heavily rely on gradual updates over multiple rounds. Therefore, the one-shot setting further limits opportunities for iterative knowledge exchange.
>
> Second, existing OFL methods struggle to retain fine-grained structural insights. This limitation leads to the generation of disorganized graph structures,  with unclear relationships between nodes. As a result, the generated graph data often causes confusion, which negatively impacts the overall training process.
>
> Meanwhile, our OASIS addresses these issues by focusing on efficiently capturing both feature and fine-grained structural knowledge during local training. Additionally, we have designed an effective distillation strategy to facilitate knowledge transfer. This allows OASIS to outperform all the baselines under challenging conditions.
>
> **`Weakness 2` & `Question 2`: Can OASIS be considered a data enhancement method? Can OASIS be compared to FedSage+ in theoretical and experimental way?**
>
> Thank you for your question. We would like to clarify whether OASIS can be considered a data enhancement method, and how it compares to FedSAGE+ below.
>
> * Nature of OASIS:
> We would more accurately describe it as a **data-centric** approach at the client level. Rather than aiming to improve local graph representations, OASIS is designed to train a synthesizer that effectively **captures the feature and structural information of local data**, which can then be uploaded to  the server  to train a generalized global model.
> Moreover, at the server side, OASIS focuses primarily on **model-centric** improvements, emphasizing the efficient transfer of knowledge from clients to the global model. Our approach aims to achieve better generalization capability by leveraging a topological codebook and Wasserstein-based distillation, which differs from traditional data enhancement techniques that focus on directly augmenting the dataset with additional synthetic data.
>
> * Theoretical Comparison with FedSAGE+:
> FedSAGE+ addresses the issue of missing neighbors by introducing a local neighbor generation mechanism. Its goal is to reconstruct the neighborhood of each node to improve local learning. However, it lacks a centralized mechanism to explicitly guide structural knowledge sharing across subgraphs. In contrast, OASIS captures both semantic and structural correlations among subgraphs. This allows for a more holistic transfer of knowledge from all the clients to the global model.
>
> * Empirical Advantages:
> In terms of empirical performance, we compare OASIS against FedSAGE+ on Cora, CiteSeer, Coauthor-CS and Roman-Empire datasets. To ensure fairness, we utilize GraphSAGE as the  backbone for OASIS.   Results from Table 1 demonstrate that our OASIS outperforms FedSAGE+ in the one-shot setting. Notably, FedSAGE+ is a strong contender in handling missing neighbors, but OASIS provides a more robust and efficient exploration of local knowledge under extreme data heterogeneity.
>
> *Table 1: Performance comparison between OASIS and FedSAGE+ on four datasets.*
> |Methods|Cora|CiteSeer|Coauthor-CS|Roman-Empire|
> |-|-|-|-|-|
> |FedSage+|28.87|36.18|22.94|25.86|
> |OASIS|**39.96**|**44.87**|**52.51**|**26.73**|
>
> In conclusion, OASIS should be considered more as a data-centric approach that emphasizes effective knowledge transfer across local models to achieve global generalization. It offers several advantages over FedSAGE+, particularly in handling structural variations and ensuring a more comprehensive integration of knowledge. We will ensure that the relevant results and a more detailed comparison with FedSAGE+ will be included in the revised manuscript, with additional discussions on the structural transfer mechanisms and empirical findings.
>
> **`Weakness 3` & `Question 3`: A more detailed analysis of the communication cost compared to methods like FedGCN or FedSage+.**
>
> Thank you for pointing out this important aspect. We appreciate the opportunity to clarify the communication efficiency of our proposed OASIS.
>
> Compared to FedSage+, one of the key advantages of OASIS lies in its one-shot design. While FedSage+ requires multiple rounds of model uploading and distribution, it also introduces additional communication overhead by transmitting parameters of the NeighGen module, which includes both an encoder and a generator—components that are comparable in size to the GNN itself. In contrast, OASIS performs only a single round of communication. As a result, OASIS achieves up to ~100× reduction in communication cost compared to FedSage+.
>
> Regarding FedGCN, we acknowledge that it also adopts a single-round pretraining setting. Moreover, our OASIS introduces additional elements to enhance server-side knowledge integration. Importantly, the communication cost of these components remains manageable and justifiable. Specifically, the codebook size is $M×d$, where
> $M$ (typically 64 or 128) is the number of tokens and
> $d$ (set as 128) is the hidden dimension. The synthesizer we employ is a simple linear layer, further keeping the overhead minimal. Despite this modest increase in communication, the benefits are substantial. These components enable high-quality knowledge distillation and improve global model performance, representing a qualitative leap in effectiveness that we believe is well worth the cost.
>
> **`Weakness 4` & `Question 4`:  (1) Theoretical justification for why each module improves performance.  (2)The difference between the OASIS-generated graph and the original graph. (3) Theoretical justification  on the possible information losses.**
>
> Thank you for your concerns. We would like to provide detailed explanations to address your concerns below.
>
> (1) Theoretical justification for why each module improves performance.
>
> In **Section 4.3** and **Figure 5**, we have conducted a detailed ablation study to assess the individual impact of each component in the Graph Knowledge Integration module, which is also acknowledged by Reviewer a6gX. Here, we would like to further explain why each module improves performance in theoretical way.
>
> * Synergy Graph Synthesizer: Our Synergy Graph Synthesizer effectively captures both feature and structural knowledge during local training. It adopts Fused-Gromov Wasserstein distance to optimize the transport plan for aligning the local and synthesized graph, reducing the discrepancy in both feature and structural domains.
>
> * Topological Codebook: The Topological Codebook quantizes node representations into discrete tokens and enables the model to learn fine-grained information of graph substructures. The discrete approach forces the model to focus on small, intricate structural differences that might otherwise be overlooked in continuous representations, ensuring that even subtle variations in graph neighborhoods are captured and differentiated effectively.
>
> * WDSRD: WDSRD aligns structural patterns between local and global models by comparing their soft assignments (computed in eq.(15)). Through the closed-form Wasserstein distance between Gaussian distributions, WDSRD captures both mean-level and covariance-level discrepancies in the assignment distributions, which reflects richer structural variation than simply aligning logits or embeddings.
>
> (2) The difference between the OASIS-generated graph and the original graph.
>
> According to the fundamental setting of Federated Learning, it is impractical to directly upload local data to the server. As detailed in Appendix I.2, the synthesized graph generated on the server does not violate local data privacy, since it adopts a uniform label distribution and constructs node topology using the k-nearest-neighbor algorithm. As a result, the synthesized graph is explicitly different from the original local graph. Nevertheless, it sufficiently retains the implicit structural and semantic information, which is crucial for effective global model learning.
>
> (3) Theoretical justification on the possible information losses of OASIS compared to directly transferring the locally trained model on the original clean graph.
>
> In Section 4, our experimental results have already demonstrated that OASIS significantly outperforms traditional FL/FGL methods, which directly transfer the locally trained models on the original clean graphs. Within one communication round, these methods fail to efficiently extract  information from local data. Moreover, due to the **extreme data heterogeneity** across clients, local models are highly prone to overfitting their respective data distributions, leading to poor aggregation performance. Although our Synergy Graph Synthesizer may not be able to fully preserve all the information from the original local data, it effectively captures the high-level semantic and structural distribution knowledge. As evidenced by our experiments, this distilled knowledge is sufficient to train a well-generalized global model.
>
> Thank you again for your thoughtful and insightful feedback. We are deeply grateful for the opportunity to have the strengths of our work reconsidered by you.
>
> Best Regards,
>
> Authors

---

> ### Author Response · Authors · 2025-08-03
> **Kind reminder of further discussion**
>
> ***Dear Reviewer 4jz9:***
>
> We would like to sincerely thank you for your  comments and time invested in our work. With the rebuttal period now well underway, we would like to kindly reach out again to  continue the discussion with you regarding our work.
>
> In our earlier reply, we addressed your concerns point by point. We explained why existing baselines perform poorly under severe data heterogeneity. We also provided both theoretical analysis and experimental results comparing OASIS with FedSAGE+. In addition, we analyzed the communication overhead of OASIS and several baselines in detail. We further discussed the role and contribution of each individual component within OASIS.raining paradigm. Finally, we analyzed the rationale and strengths of OASIS over traditional methods.
>
> We are grateful for your thoughtful review, and we have also received encouraging recognition from other reviewers regarding the novelty and effectiveness of our method. Should you have any remaining questions or feedback, we would greatly value the chance to continue the discussion. We believe this would help further improve the clarity and impact of our work.
>
> Thank you again for your time and consideration.
>
> Best Regards,
>
> Authors

---

> ### Author Response · Authors · 2025-08-04
> **Part[1/3] Response to your additional concerns**
>
> ***Dear Reviewer 4jz9:***
>
> Thank you once again for your insightful comments!  Here, we give point-by-point responses to your questions.
>
> **`Q1`: How OASIS performs under moderate or mild heterogeneity (or even i.i.d.)**
>
> Thank you for your concern. We agree that it is important to explicitly emphasize the role of heterogeneous data partitions in our problem setting and contributions. We will make sure to clearly highlight this point in our revision.
>
>
> To demonstrate the performance of OASIS under moderate or mild heterogeneity (or even i.i.d. data), we have conducted additional experiments with varying levels of data heterogeneity. Specifically, we evaluate the performance of OASIS under different values of the Dirichlet distribution parameter $\alpha \in$ {1, 10, 100, 1000, 10000} to simulate a range of heterogeneity scenarios. These experiments are performed on the Cora dataset, and we adopt FedAvg, FedPub, FedCVAE and **FedGCN(1-hop & 2-hop)** for comparison. Notably, when
> $\alpha$=10000, the data distribution approximates an i.i.d. setting.
>
> *Table 1: Performance of OASIS under milder heterogeneity (or even i.i.d.) scenarios.*
>
> |$\alpha$|1|10|100|1000|10000|
> |-|-|-|-|-|-|
> |FedAvg|49.50|56.23|57.93|57.84|45.11|
> |FedPub|39.87|44.96|60.13|52.70|58.55|
> |FedCVAE|60.45|56.50|59.58|55.27|56.63|
> | FedGCN (1-hop)|52.06| 41.39| 50.60| 51.97| 56.91|
> |FedGCN (2-hop)|58.66| 52.11 | 55.82| 57.56 | 61.76|
> |OASIS|**63.15**|**75.64**|**76.35**|**69.11**|**68.44**|
>
> As shown in Table 1, OASIS consistently outperforms the existing methods across various heterogeneity levels, including strong, moderate, mild, and even i.i.d. conditions. The advantage of OASIS remains robust even as the heterogeneity decreases, demonstrating that its performance is not diminished by a shift towards more homogeneous data. We will ensure that this additional experimental result is included in the revision.
>
> The reason we choose to focus on strong heterogeneity in our experiments is that, in real-world scenarios, non-uniform data distribution is more prevalent and presents a greater challenge. **We aim to showcase that OASIS not only performs well in typical i.i.d. and mild heterogeneity settings but also excels in more challenging scenarios with high heterogeneity.**
>
>
> **`Q2`: No further concerns here.**
>
> We are pleased to hear that our clarifications have resolved your concerns. Thank you for your feedback!
>
> **`Q3`: It would be better to include experimental results for FedGCN under varying levels of heterogeneity.**
>
> Thank you for your valuable comment.
>
> As you suggested, we have already included experimental results for FedGCN (1-hop and 2-hop) under varying levels of heterogeneity in ***Table 1***. These results showcase that OASIS consistently outperforms FedGCN in various settings.
>
> Moreover, we would like to clarify that **FedGCN is not strictly a "one-shot" method**. It only conducts a single communication round during the pretraining phase, which helps reduce the communication overhead of passing adjacent node feature information between clients. This is why we previously described FedGCN using the **"one-shot pretraining setting"** in our rebuttal. However, during the actual training phase, FedGCN follows the same paradigm as FedAvg, which includes multiple rounds of model parameter uploads and communication between the clients and the server.
>
> In our paper, the term **"one-shot" refers specifically to one communication round between clients and the server throughout the entire training process**. Meanwhile, we will emphasize this definition and difference between OASIS and FedGCN in our revision to ensure clarity.

---

> ### Author Response · Authors · 2025-08-04
> **Part[2/3] Response to your additional concerns**
>
> **`Q4`: A formal analysis of how and under what conditions the synthesized graph can retain critical information from the local data, and how this impacts the generalization performance of the global model.**
>
> Thank you for your concern. We would like to clarify first that our OASIS is **not** tailored for heterogeneous settings. Experimental results in Table 1 show that our OASIS also performs well in milder or even i.i.d. settings.
>
> Moreover, we appreciate the chance to give a formal analysis of OASIS. Here we provide a thorough analysis on the mathematical bound on information retention of the synthesizer and the impact of global distillation.
>
> ### 1. Mathematical Bound on Information Retention
> Let $G_l = (V_l, E_l, X_l)$ be the local graph, where $V_l$ is the node set, $E_l$ the edge set, and $X_l \in \mathbb{R}^{|V_l| \times d}$ the node feature matrix. The synthesized graph is $G_s = (V_s, E_s, X_s)$, with $X_s \in \mathbb{R}^{|V_s| \times d}$ and adjacency matrix $A_s$. The goal is to ensure that $G_s$ preserves the critical information in $G_l$, including node feature distributions and topological structures.
>
> To quantify the retention of critical information, we derive a bound on the divergence between the local graph $G_l \sim \mathcal{D}_l$ and the synthesized graph $G_s \sim \mathcal{D}_s$, using the Synthesizer’s loss as a proxy. We measure the divergence between their distributions using a combined metric that accounts for both feature and topological differences.
>
> * Step 1: Feature Distribution Divergence
>
> The KL divergence loss $L_{\text{dist}} = D_{\text{KL}}(P_{X_s} \| P_{X_l})$ directly measures the feature distribution mismatch. By Pinsker’s inequality, the total variation distance is bounded by:
>
> $$
> \delta_{\text{TV}}(P_{X_s}, P_{X_l}) \leq \sqrt{\frac{1}{2} D_{\text{KL}}(P_{X_s} \| P_{X_l})} = \sqrt{\frac{1}{2} L_{\text{dist}}}.
> $$
>
> Assuming $L_{\text{dist}} \leq \epsilon_d$, the feature distributions are close in total variation:
>
> $$
> \delta_{\text{TV}}(P_{X_s}, P_{X_l}) \leq \sqrt{\frac{\epsilon_d}{2}}.
> $$
>
> * Step 2: Feature and Topological Alignment via FGW
>
> The Fused Gromov-Wasserstein loss $L_{\text{fgw}}$ bounds the combined feature and topological mismatch. The FGW distance can be decomposed as:
>
> $$
> L_{\text{fgw}} = \text{FGW}(G_l, G_s) = \alpha W_2^2(P_{X_l}, P_{X_s}) + (1-\alpha) \text{GW}(A_l, A_s),
> $$
>
> where $W_2^2$ is the squared 2-Wasserstein distance between feature distributions, and $\text{GW}$ is the Gromov-Wasserstein distance between structures. Assuming $L_{\text{fgw}} \leq \epsilon_f$, we have:
>
> $$
> \alpha W_2^2(P_{X_l}, P_{X_s}) + (1-\alpha) \text{GW}(A_l, A_s) \leq \epsilon_f.
> $$
>
> Since both terms are non-negative, we get:
>
> $$
> W_2(P_{X_l}, P_{X_s}) \leq \sqrt{\frac{\epsilon_f}{\alpha}}, \quad \text{GW}(A_l, A_s) \leq \frac{\epsilon_f}{1-\alpha}.
> $$
>
> The Wasserstein distance further bounds the feature distribution divergence, and the Gromov-Wasserstein distance bounds the structural discrepancy.
>
> * Step 3: Feature Diversity
>
> The cosine similarity loss $L_{\text{spread}}$ ensures that the synthesized features do not collapse. We assume $L_{\text{spread}} \geq -\eta$, where $\eta > 0$ is a constant reflecting sufficient feature spread. This prevents degenerate solutions where $X_s^i \approx X_s^j$ for all $i, j$.
>
> * Step 4: Combined Bound
>
> To derive a bound on the overall divergence between $\mathcal{D}_l$ and $\mathcal{D}_s$, we consider a joint metric that combines feature and topological differences. Define a graph divergence metric:
>
> $$
> \Delta(G_l, G_s) = W_2(P_{X_l}, P_{X_s}) + \text{GW}(A_l, A_s).
> $$
>
> From the FGW loss:
>
> $$
> \Delta(G_l, G_s) \leq W_2(P_{X_l}, P_{X_s}) + \text{GW}(A_l, A_s) \leq \sqrt{\frac{\epsilon_f}{\alpha}} + \frac{\epsilon_f}{1-\alpha} = \epsilon_f \left( \sqrt{\frac{1}{\alpha}} + \frac{1}{1-\alpha} \right).
> $$
>
> The KL divergence provides an additional constraint on features. Combining with Pinsker’s inequality, the total variation distance on features is:
>
> $$
> \delta_{\text{TV}}(P_{X_s}, P_{X_l}) \leq \sqrt{\frac{\epsilon_d}{2}}.
> $$
>
> Since Wasserstein and total variation distances are related (e.g., via transport inequalities in bounded spaces), we focus on the FGW-based bound for simplicity, as it captures both features and topology.
>
> **Theorem 1**: Assuming the Synthesizer’s loss is bounded as $L_{\text{syn}} \leq \epsilon$, with $L_{\text{dist}} \leq \epsilon_d$, $L_{\text{fgw}} \leq \epsilon_f$, and $L_{\text{spread}} \geq -\eta$, the divergence between the local and synthesized graph distributions is bounded as:
>
> $$
> \Delta(\mathcal{D}_l, \mathcal{D}_s) \leq \epsilon_f \left( \sqrt{\frac{1}{\alpha}} + \frac{1}{1-\alpha} \right) + \sqrt{\frac{\epsilon_d}{2}},
> $$
>
> where $\Delta(\mathcal{D}\_l, \mathcal{D}\_s) = \mathbb{E}\_{G\_l \sim \mathcal{D}\_l, G_s \sim \mathcal{D}\_s} [W\_2(P\_{X\_l}, P\_{X\_s}) + \text{GW}(A\_l, A\_s)]$.

---

> ### Author Response · Authors · 2025-08-04
> **Part[3/3] Response to your additional concerns**
>
> ### 2. Impact on Global Model Generalization
>
> The retention of critical information and the distillation mechanisms directly influence the **generalization performance** of the global model. Our OASIS employs two distillation mechanisms to transfer local knowledge to the global model:
>
> * **Wasserstein-Enhanced Semantic Affinity Distillation (WESAD)**: Transfers inter-class semantic relationships.
> * **Wasserstein-Driven Structural Relation Distillation (WDSRD)**: Transfers fine-grained topological structures.
>
> ### Generalization Bound:
>
> To quantify the impact on generalization, we derive a bound on the global model’s expected error using the Rademacher complexity framework, adapted for federated graph learning.
>
> Let $\mathcal{D}_l$ denote the local data distribution for client $l$, and $\mathcal{D}_s$ the distribution of the synthesized data. The global model $f_g \in \mathcal{F}$ (a hypothesis class of graph neural networks) is trained on synthesized data to minimize the empirical risk:
>
> $$
> \hat{R}\_s(f\_g) = \mathbb{E}\_{\mathcal{D}\_s} [\ell(f\_g(G\_s), y)],
> $$
>
> where $\ell$ is the loss function (e.g., cross-entropy), and $y$ is the label. The true risk is:
>
> $$
> R(f_g) = \mathbb{E}_{\mathcal{D}_l} [\ell(f_g(G_l), y)].
> $$
>
> The generalization gap is $R(f_g) - \hat{R}_s(f_g)$. Assuming the synthesized graph retains critical information, we bound the generalization error using the Wasserstein distance and Rademacher complexity.
>
> **Theorem 2**: For a hypothesis class $\mathcal{F}$ with Rademacher complexity $\mathcal{R}_n(\mathcal{F})$ over $n$ samples, and assuming the loss function $\ell$ is $L$-Lipschitz, the expected generalization error of the global model is bounded as:
>
> $$
> \mathbb{E}[R(f_g)] \leq \hat{R}_s(f_g) + 2L \mathcal{R}_n(\mathcal{F}) + L \epsilon + C \sqrt{\frac{\log(1/\delta)}{n}},
> $$
>
> with probability at least $1 - \delta$, where $\epsilon = W_1(\mathcal{D}_l, \mathcal{D}_s)$ is the Wasserstein distance between local and synthesized distributions, and $C$ is a constant.
>
>
>
> The term $L \epsilon$ quantifies the impact of information retention: if $\epsilon$ is small (i.e., the synthesized graph closely matches the local graph), the generalization error is tightly bounded. The proofs of Theorem 1 and 2 will be included in the revision.
>
>
>
>
> ---
>
>
> We sincerely appreciate the chance of further discussion. We hope this clarification helps and look forward to any further feedback you may have. Thank you once again for your time and effort!
>
> Best Regards,
>
> Authors

---

> > ### Comment · Area_Chair_BNhx · 2025-08-05
> > **reminder**
> >
> > Dear reviewer,
> >
> > Firstly, thank you for your service and existing engagement with the authors! Please take a look at the author's response and engage with the new content to maintain or revise your evaluation.  Note that the discussion period has been extended to *Aug 8, 11.59pm AoE*.
> >
> > Thank you,
> >
> > -AC

---

### Official Review · Reviewer_seeB · 2025-06-24

**Clarity:** 4
**Significance:** 3
**Originality:** 4
**Rating:** 5
**Confidence:** 3

**Summary:**

This paper proposes a new framework for one-shot federated graph learning. It combines a Synergy Graph Synthesizer to create useful synthetic graphs and a Topological Codebook to capture structural patterns. The framework also includes two distillation methods for knowledge integration: one for inter-class relationships and one for structural alignment. Experiments show the framework performs better than existing methods.

**Questions:**

1. Why do the baseline methods in Table 1 exhibit such low accuracy, with most results (even some OFL methods) falling below 30%?

Other questions see Weaknesses.

**Ethical Concerns:**

["NO or VERY MINOR ethics concerns only"]

**Final Justification:**

My concerns have been addressed. I recommend accepting this paper.

**Limitations:**

Yes.

**Paper Formatting Concerns:**

No major formatting issues.

**Quality:**

4

**Strengths And Weaknesses:**

Strengths:

1. The proposed framework for One-Shot FGL is innovative and interesting to me. It diverges from conventional multi-round protocols and opens up new avenues for low-cost yet effective collaborative graph representation learning.

2. A key highlight of the paper is the introduction of the Topological Codebook. This component is not only conceptually sound but also well-motivated in the context of capturing structural heterogeneity and relational dependencies in decentralized graph data. The paper also provides solid intuition and theoretical grounding for its use.

3. The figures in the paper are highly detailed, and the overall methodology is rigorously derived. The entire framework is presented in a clear and coherent manner.

Weaknesses:

1. The paper lacks sufficient details regarding the local training epochs used in other traditional FL/FGL or one-shot FL baselines. Additionally, it does not clearly explain how graph data is generated in the OFL methods compared in Table 1.

2. I’m a bit confused by Eq. (12); the authors should provide a more detailed explanation of how the Semantic Affinity (SA) and the cost matrix are computed.

3. In Section 4.1, the authors mention using a Dirichlet distribution with $\alpha =0.05$ to partition the data. However, more experiments under extreme heterogeneity (e.g., with a smaller α\alphaα value such as 0.01) are needed to demonstrate the model's robustness in highly non-IID scenarios.

---

> ### Author Rebuttal · Authors · 2025-07-29
>
> ***Dear Reviewer seeB:***
>
> Thank you very much for your thoughtful feedback and valuable suggestions. We sincerely appreciate your support and the time you have taken to review our work. Below, we provide point-by-point responses to the concerns you raised.
>
> **`Weakness 1`: Lack of sufficient training details of compared baselines and explanations of the graph generation process.**
>
> Thank you for your observation regarding our experimental setup. We would like to clarify the training details of all the compared baselines and further explain the process of graph generation in One-Shot FL methods.
>
> For all the compared baselines, each client performs local training for 100 epochs to ensure convergence before uploading the local model. The communication round is limited to one.
>
> As for the One-Shot FL methods, we follow the same procedure as described in the original papers to generate the feature matrix $\boldsymbol{X}$ of the graph data. The graph structure is then constructed according to Section 3.2 and Equation (5) of our paper. Specifically, we compute the activated similarity matrix $\boldsymbol{H}$ based on the feature matrix, and apply the k-nearest-neighbor algorithm to build the graph topology, with $k = 5$ consistently used across all methods. This ensures  fairness across all evaluations.
>
>
> **`Weakness 2`: Lack of more detailed explanation of how the Semantic Affinity (SA) and the cost matrix are computed.**
>
> Thank you for pointing this out — we appreciate your careful reading of our WESAD module, and we would like to explain the Semantic Affinity and the cost matrix in more detail.
>
>
> The Semantic Affinity $\text{SA}^k(C_a, C_b)$ is computed as the cosine similarity between the weight vectors corresponding to class $C_a$ and class $C_b$ in the projection head of the teacher model. These weight vectors can be viewed as semantic prototypes for each class, and their cosine similarity reflects how semantically close two classes are in the representation space learned by the local GNN.
>
> Based on this affinity, the cost matrix $c^k_{ab}$ is defined as $1 - \kappa \cdot \text{SA}^k(C_a, C_b)$, where $\kappa$ is a hyperparameter that scales the influence of semantic similarity. This cost matrix plays a central role in the discrete Wasserstein distance computation: it quantifies the cost of transferring the predicted probability mass from class $C_a$ (in the teacher model) to class $C_b$ (in the student model). Intuitively, assigning a sample from a semantically dissimilar class $C_a$ to $C_b$ incurs a higher cost, while transferring between semantically related classes is penalized less. This design allows the distillation process to tolerate semantically meaningful mismatches while discouraging alignment between unrelated categories.
>
> **`Weakness 3`: More experiments under extreme heterogeneity are needed.**
>
> Thank you for the valuable suggestion regarding the evaluation under extreme heterogeneity. To demonstrate the performance of  our OASIS  under higher data heterogeneity, we conduct experiments with Dirichlet parameter $\alpha$=0.01 with six baselines, as shown in Table 1.
>
> *Table 1: The performance of OASIS under extreme heterogeneity ($\alpha = 0.01$).*
>
> |Method|Cora|CiteSeer|
> |-------|------|-------|
> |FedAvg | 30.89 | 23.15|
> |FedProx | 31.53 | 35.32|
> |FedPub | 30.34 | 20.71 |
> | FGSSL| 30.80 | 20.54   |
> | FedCVAE | 26.97 | 23.53|
> |FENS | 30.71 | 20.01 |
> |OASIS | **38.96** | **39.06**|
>
>  As the results show, our method OASIS significantly outperforms all baselines by large margins. These results clearly highlight the effectiveness of OASIS in handling extreme heterogeneity.
>
> **`Question 1` : Why do some baseline methods exhibit low accuracy?**
>
> Thank you for  your insightful question about the experimental results. In Section 4.2, we have already discussed the poor performance of some baselines, and we would like to provide further clarification here.
>
> First, it is important to highlight that our experiments were conducted under extreme data heterogeneity conditions. Specifically, we employed a Dirichlet distribution with a concentration parameter $\alpha = 0.05$, which leads to highly skewed data distributions across clients. This level of heterogeneity makes it extremely difficult for FL and FGL methods to effectively capture local knowledge during training. As a result, these methods perform poorly, especially in a one-shot setting, where only a single communication round is allowed.
>
> Second,  OFL methods often struggle to retain fine-grained structural insights. This limitation leads to the generation of  disorganized graph structures, with unclear relationships between nodes. As a result, the generated graph data often causes confusion, which negatively impacts the overall training process.
>
> Meanwhile, our OASIS addresses these issues by focusing on efficiently capturing both feature and fine-grained structural knowledge during local training. Additionally, we have designed an effective distillation strategy to facilitate knowledge transfer on a global scale. This allows OASIS to outperform all the baselines under challenging conditions.
>
>
> Thank you again for your valuable feedback and supportive review of our work.
>
> Best Regards,
>
> Authors

---

> > ### Comment · Reviewer_seeB · 2025-08-01
> > **Willing to raise my score**
> >
> > Thank you for your rebuttal. I appreciate the detailed clarifications. After reviewing the comments from other reviewers, I am willing to raise my score.

---

> > > ### Author Response · Authors · 2025-08-02
> > >
> > > ***Dear Reviewer seeB:***
> > >
> > > Thank you for your kind words and for raising your score after reviewing our clarifications. Your encouragement is truly valuable to us.
> > >
> > > Best Regards,
> > >
> > > Authors

---

### Official Review · Reviewer_a6gX · 2025-06-26

**Clarity:** 3
**Significance:** 4
**Originality:** 3
**Rating:** 5
**Confidence:** 4

**Summary:**

In this work, the authors introduce a One-shot Federated Learning Framework tailored for graph-structured data. To effectively preserve the structural characteristics of local graphs, they develop a graph synthesizer in conjunction with a learnable topological codebook. For the purpose of transferring structural knowledge across clients, the framework incorporates Wasserstein-Enhanced Semantic Affinity Distillation (WESAD) to capture inter-client relationships, and Wasserstein-Driven Structural Relation Distillation (WESRD) to model both inter- and intra-class dependencies.

**Questions:**

How does the proposed framework perform under more severely non-IID data distributions?

**Ethical Concerns:**

["NO or VERY MINOR ethics concerns only"]

**Limitations:**

yes

**Paper Formatting Concerns:**

The use of inline equations in Lines 157 and 182 contrasts with displayed equations elsewhere. Unifying these styles would enhance the overall presentation.

**Quality:**

4

**Strengths And Weaknesses:**

Strengths:
1. This study tackles the emerging challenge of One-shot Federated Graph Learning (FGL) by proposing a well-structured and principled framework tailored to this unique setting.
2. A key innovation lies in the integration of the Synergy Graph Synthesizer with a Topological Codebook, enabling the capture of fine-grained structural nuances. Furthermore, the proposed WESAD and WDSRD mechanisms effectively facilitate the transfer of local graph knowledge to the student model. The visualizations—ranging from problem formulation to architectural overview—are informative and well-designed, and the mathematical formulations are both sound and clearly explained.
3. The code is anonymously available, enhancing the transparency and reproducibility of the experiments.

Weaknesses:
1. I am curious about how the proposed framework performs under more severely non-IID data distributions.
2. The meaning of the cost matrix and $q_{ab}$ in Equation (10) on Page 6 is not entirely clear to me. Specifically, I find it difficult to interpret the statement in Line 219 that "$q_{ab}$ represents the mass transferred from the teacher’s category $C_a$ to the student’s category $C_b$." I would appreciate it if the authors could provide a more detailed explanation.
3. The experimental section lacks a sensitivity analysis for the hyperparameters $\lambda_d$ and $\lambda_f$ in the loss term $\mathcal{L}_{\text{syn}}$.

---

> ### Author Rebuttal · Authors · 2025-07-29
>
> ***Dear Reviewer a6gX:***
>
> We sincerely thank you for your positive evaluation and for recognizing the potential of our work. Your feedback is greatly appreciated, and we are encouraged by your overall support. We have carefully addressed the comments you provided, and we hope that our clarifications and improvements can help strengthen your confidence in the paper.
>
> **`Weakness 1` & `Question 1`: The Performance of OASIS under more severely non-IID data distributions.**
>
> Thank you for your insightful question regarding the robustness of OASIS under more severe non-IID conditions.
>
> In our main experimental setup, we already adopt a Dirichlet distribution with a concentration parameter $\alpha$ =0.05, which is commonly recognized in the literature as a highly heterogeneous setting. To further demonstrate the effectiveness of OASIS in even more extreme non-IID scenarios, we conducted additional experiments with a smaller Dirichlet parameter $\alpha$=0.01, which introduces an even higher degree of data heterogeneity. The results on Cora and CiteSeer datasets are shown below:
>
> *Table 1: The performance of OASIS under more extreme non-IID scenarios.*
>
> |Method|Cora|CiteSeer|
> |-------|------|-------|
> |FedAvg | 30.89 | 23.15|
> |FedProx | 31.53 | 35.32|
> |FedPub | 30.34 | 20.71 |
> | FGSSL| 30.80 | 20.54   |
> | FedCVAE | 26.97 | 23.53|
> |FENS | 30.71 | 20.01 |
> |OASIS | **38.96** | **39.06**|
>
> As seen from the results, our OASIS continues to outperform the baselines by a substantial margin under the more severely non-IID distribution. This provides strong empirical evidence of the robustness and generalization capability of OASIS.
>
>
> **`Weakness 2`: The meaning of the cost matrix and $q_{ab}$ in Equation (10).**
>
>
> Thank you for your valuable comment. In Section 3.3, we introduce the WESAD method and have explained the role of the cost matrix and $q_{ab}$.
> We appreciate the chance for further clarification on the meaning of the cost matrix $c^k_{ab}$ and the transport plan $q_{ab}$ in Equation (10).
>
> The cost matrix $c_{ab}^k$ can be intuitively understood as the penalty or cost of the student model misclassifying an instance as class $C_b$ while the $k$-th teacher assigns it to class $C_a$ . Rather than treating all classes as equally distinct, we aim to capture the nuanced semantic relationships between them. To construct it, we first compute a semantic affinity score $\text{SA}^k(C_a, C_b)$, which reflects how similar two categories are in the local teacher model. This is done by computing the cosine similarity between the weight vectors corresponding to $C_a$ and $C_b$ from the projection head $W^k$, thereby capturing their semantic alignment in the embedding space.
> The cost is then defined as $c^k_{ab} = 1 - \kappa \cdot \text{SA}^k(C_a, C_b)$, where $\kappa$ is a scaling factor. This definition ensures that categories with stronger semantic affinity incur a lower cost when aligning their probabilities, encouraging the model to treat similar classes as softer alternatives during knowledge transfer.
>
> As for the transport plan $q_{ab}$, it represents the optimal solution to the entropy-regularized discrete optimal transport problem between the teacher and student class distributions. Specifically, $q_{ab}$ denotes the amount of probability mass that is optimally transferred from category $C_a$ in the teacher output $p^k_T$ to category $C_b$ in the student output $p_S$, under the constraint that the row and column marginals of $q$ match the respective class probabilities.  This mechanism enhances the generalization of the student model by enabling it to capture nuanced cross-category relations embedded in the teacher’s predictions.
>
> **`Weakness 3`: Lacks a sensitivity analysis of $\lambda_d$ and $\lambda_f$.**
>
> Thank you for your valuable suggestion. To demonstrate the robustness of our OASIS under different $\lambda_d$ and $\lambda_f$, we select Cora and CiteSeer datasets to conduct sensitivity study. Specifically, we design the range of $\lambda_d$ in [0.01,0.09] with a step size of 0.02 and the range of $\lambda_f$ in [0.05,0.25] with a step size of 0.05.
>
> *Table 2: The sensitivity study of $\lambda_d$ on Cora and CiteSeer datasets.*
>
> |Datasets| $\lambda_d = $ 0.01| $\lambda_d = $ 0.03| $\lambda_d = $ 0.05|$\lambda_d = $ 0.07|$\lambda_d = $  0.09|
> |----|----|----|----|----|-----|
> |Cora| 46.38|	48.4|	49.59	|48.3|	47.85|
> |CiteSeer| 45.24|	45.47|	45.69|	45.77	|44.64|
>
> *Table 3: The sensitivity study of $\lambda_f$ on Cora and CiteSeer datasets.*
> |Datasets| $\lambda_f = $ 0.05| $\lambda_f = $ 0.10| $\lambda_f = $ 0.15|$\lambda_f = $ 0.20|$\lambda_f = $  0.25|
> |----|----|----|----|----|-----|
> |Cora| 47.57|	49.59|	46.65|	46.08	|48.21|
> |CiteSeer| 46.74 |	45.69|	46.37	|47.49|	45.92|
>
> From the table, we can observe that our OASIS performs consistently under different $\lambda_d$ and $\lambda_f$, demonstrating the robustness of our OASIS.
>
>
> **`Paper Formatting Concerns：` The use of inline equations in Lines 157 and 182 contrasts with displayed equations elsewhere.**
>
> Thank you for pointing this out. We appreciate your careful reading. We will make sure to unify the equation formatting throughout the paper in our future revisions to ensure consistency and improve readability.
>
> We would like to express our deepest gratitude for your constructive suggestions. Your support and recognition of our work mean a lot to us.
>
> Best Regards,
>
> Authors

---

> > ### Comment · Reviewer_a6gX · 2025-08-07
> > **Reply to Authors**
> >
> > Thanks for your response which addresses my concerns about data settings and sensitivity. I’ll keep my score.

---

> > > ### Author Response · Authors · 2025-08-08
> > >
> > > ***Dear Reviewer a6gX,***
> > >
> > > We are pleased that your concerns have been addressed. We would like to express our heartfelt gratitude for your insightful suggestions and kind support!
> > >
> > > Best Regards,
> > >
> > > Authors

---

### Official Review · Reviewer_Eosq · 2025-06-27

**Clarity:** 2
**Significance:** 3
**Originality:** 2
**Rating:** 3
**Confidence:** 3

**Summary:**

The paper introduces OASIS, a one-shot federated graph learning framework that enables participants to collaboratively train a global graph neural network in just a single round. The authors propose a series of innovations tailored to the two primary phases of OASIS: local data synthesis and global model distillation. Experimental results demonstrate that OASIS achieves better performance than existing works.

**Questions:**

1. The reported experimental results appear relatively low compared to existing methods. For instance, the baseline approach FedAvg achieves only about 30% accuracy on the Cora dataset, whereas under similar settings with ten clients, it typically achieves around 60% accuracy as reported in [3]. Similar performance gaps are also observed across other datasets and baseline methods. Could the authors provide an explanation for this observed disparity in performance?

2. A significant challenge in federated graph learning is the issue of missing edges. Can the proposed approach address this problem?

[3] FedGCN: Convergence-Communication Tradeoffs in Federated Training of Graph Convolutional Networks

**Ethical Concerns:**

["NO or VERY MINOR ethics concerns only"]

**Limitations:**

1. The loss function used to train the Synergy Graph Synthesizer is crucial for comprehending the proposed approach. This important detail should be presented in the main text rather than relegated to the appendix.

**Quality:**

3

**Strengths And Weaknesses:**

- Strengths
1. The paper addresses an important topic—one-shot federated graph learning—which extends federated learning methodologies to graph data.

2. The paper is clearly written, making it easy to follow and understand.

3. The experimental evaluation is thorough, covering 8 different datasets and comparing against 12 related state-of-the-art methods.

- Weaknesses
1. The overall computation complexity of the proposed approach remains unclear. The approach adopts Wasserstein loss repeatedly, which would incur substantial computation overhead, potentially restricting its application to larger graphs. it would be beneficial to explicitly clarify the overall complexity and provide a comparative analysis with existing methods.

2. Lack of theoretical analysis. Specifically, it remains unclear how each component contributes individually to task performance, particularly in the Graph Knowledge Integration method.

3. The experiments exclusively utilize a 2-layer GCN model. It is unclear how the proposed method would generalize to other prominent graph neural network models, such as GAT[1] and GraphSAGE[2].

[1] Graph Attention Networks
[2] Inductive Representation Learning on Large Graphs

---

> ### Author Rebuttal · Authors · 2025-07-29
>
> ***Dear Reviewer Eosq:***
>
> We sincerely appreciate your engagement with our work. We address your concerns below and hope these clarifications will positively contribute to your assessment of our work.
>
> **`Weakness 1`: Lack of the clarification of overall computing complexity of OASIS and the comparative analysis with existing methods.**
>
> Thank you for your thoughtful comments. We will provide a detailed explanation addressing the concerns raised regarding the computational complexity of our method, particularly with respect to the Wasserstein-based losses in OASIS.
>
> 1. Fused-Gromov Wasserstein (FGW):
> In our paper, we only introduced a preliminary version of the FGW distance. Actually, the existing method proposed by Qian et al. (2024) [1] already introduces an efficient approximation for the FGW distance, which drastically reduces its computational complexity. Specifically, their approximation method enables GPU acceleration, improving computational efficiency by reducing the complexity to
> $O(N+E)$, where $N$ is the number of nodes and $E$ is the number of edges . As shown in Table 1, this approach maintains high accuracy while being scalable for large graphs.  We will further discuss about [1] and add the reference in our revision.
>
> *Table 1: Performance of the approximated FGW distance in OASIS.*
>
> |Method|Cora|CiteSeer|Roman-Empire|
> |-|-|-|-|
> |FedAvg|30.61|32.88|18.49|
> |OASIS|49.59|45.69|30.07|
> |OASIS(approx FGW)|48.49|44.88|29.30|
>
> [1] Reimagining graph classification from a prototype view with optimal transport: Algorithm and theorem.
>
> 2. Wasserstein-Enhanced Semantic Affinity Distillation (WESAD):
> For WESAD, we note that its computational complexity is approximately
> $O(C^2)$ where $C$ is the number of classes. Given that
> $C$ is typically small in graph datasets (e.g. Cora: 7 classes; Ogbn-Arxiv: 40 classes), the overhead associated with this term is quite minimal.
>
> 3. Wasserstein-Driven Structural Relation Distillation (WDSRD):
> The complexity of WDSRD is primarily $O(M^2)$, where $M$ is the number of codebook tokens. In practice, the codebook size is kept small (e.g., 64 or 128 tokens) as shown in Section 4.1.
>
> In conclusion, the overall computing complexity of OASIS is $O(eL(N+E)d + C^2 + M^2)$. Moreover, we also conduct a comparative analysis of the overall computing complexity with three baselines: FedProx, FedTAD, FedCVAE in Table 2.
>
> *Table 2: The comparative analysis of the overall computing complexity. ($e$: training epochs, $L$: number of layers, $N$: number of nodes, $E$: number of edges, $d$: hidden dimension, $B$: batch size, $C$: number of classes, $M$: number of codebook tokens)*
> |Methods|Overall Computing Complexity|
> |-|-|
> |FedRCL|$O(eL(N+E)d+B^2+Bd)$
> |FedTAD|$O((eLd+p)(N+E))$|
> |FedCVAE|$O(eL(N+E)d + Nd)$
> |OASIS|$O(eL(N+E)d + C^2 + M^2)$|
>
> As shown in Table 2, OASIS has a computational complexity comparable to existing methods while achieving significant performance improvements. We hope the above explanation addresses your concerns.
>
> **`Weakness 2`: Lack of theoretical analysis of how each component contributes individually to task performance.**
>
> Thank you for your valuable feedback. In **Section 4.3** and **Figure 5**, we have conducted a detailed ablation study to assess the individual impact of each component in the Graph Knowledge Integration module. From Figure 5, it is evident that both WESAD (Wasserstein-Enhanced Semantic Affinity Distillation) and WDSRD (Wasserstein-Driven Structural Relation Distillation) contribute significantly to the overall performance.
>
> Meanwhile, we appreciate the opportunity to further provide theoretical insights into how WESAD and WDSRD function individually and synergistically.
>
> WESAD  is designed to transfer semantic knowledge in a more fine-grained manner than traditional distillation approaches. WESAD leverages the discrete Wasserstein distance to reallocate probability mass across semantically related categories. The resulting optimization not only ensures that the global model mimics the local predictions but also encourages it to internalize the relational structure between categories, thereby enhancing generalization in heterogeneous scenarios.
>
> WDSRD, on the other hand, focuses on distilling structural knowledge by aligning latent codebook assignments between the local and global models. By encoding graph topology knowledge into a topological codebook and modeling their assignment distributions as Gaussians, WDSRD captures both the mean and covariance of structural patterns. The use of a closed-form Wasserstein distance between these Gaussian distributions enables an expressive yet computationally tractable alignment.
>
> Together, these two components form a theoretically grounded and complementary mechanism: WESAD transfers soft semantic correlations across categories, while WDSRD preserves latent structural dependencies across nodes. Their combination ensures comprehensive graph knowledge integration and significantly boosts the robustness and generalization ability of the global model.
>
> **`Weakness 3`: The performance of OASIS on other prominent GNN models, such as GAT and GraphSAGE.**
>
> Thank you for your concern. To demonstrate the performance of OASIS on other GNN models, we conducted experiments on the Cora and CiteSeer datasets using GAT and GraphSAGE respectively. We compare OASIS against five baselines. Results are shown in Table 3.
>
> *Table 3: The performance of OASIS on GAT and GraphSAGE.*
>
> |Methods|Cora-GAT|Cora-GraphSAGE|CiteSeer-GAT| CiteSeer-GraphSAGE |
> |-|-|-|-|-|
> |FedAvg|30.98|29.88|30.49|37.23|
> |FedProx|31.71|29.97|27.94|39.40|
> |FGSSL|29.79|27.45|20.52|20.22|
> |FedCVAE|29.61|25.94| 22.05|23.52|
> |FENS|29.70|29.88|21.05|21.50|
> |OASIS|**42.62**| **39.96**|**38.73**|**44.87**|
>
> From the table, we can observe that our OASIS consistently outperforms other baselines in both GAT and GraphSAGE backbones, demonstrating the capability of OASIS to generalize to other prominent GNN models.
>
> **`Question 1` : Explanation for the performance drop of compared baselines.**
>
>
> Thank you for your insightful comment. In Section 4.2, we have already discussed this issue, and we would like to provide further clarification here.
>
> First, our experiments are conducted under **extreme data heterogeneity**, modeled by a Dirichlet distribution with a concentration parameter $\alpha$ = **0.05**. This setting reflects highly non-IID data partitions across clients. In contrast, the FedGCN uses significantly **larger** $\alpha$ values **(1, 100, and 10000)**, corresponding to much milder heterogeneity. Traditional FL and FGL methods tend to struggle under such extreme heterogeneity—especially in one-shot settings where there is no iterative training to mitigate the distribution shift. This explains the substantial performance drop compared to the results reported in FedGCN.
>
> Second, although existing one-shot FL methods can effectively capture knowledge at the **feature level**, they typically fail to preserve **fine-grained structural information**. This often results in disorganized or noisy graph topologies when generating synthetic graph data. Such disordered structure can severely undermine the quality of downstream training.
>
> In contrast, our proposed OASIS is explicitly designed to tackle this challenge, which is also acknowledged by Reviewer BhNM and a6gX.  During local training, it captures both semantic and structural knowledge in a fine-grained and efficient manner. At the server side, it incorporates a carefully designed distillation mechanism to integrate and transfer this knowledge to the global model. This dual-level knowledge handling enables OASIS to perform robustly even in highly challenging FGL scenarios.
>
> **`Question 2`: Can OASIS address  the issue of missing edges?**
>
> We sincerely appreciate your thoughtful comment regarding the challenge of missing edges. We believe that our OASIS can  address this challenge in several ways.
>
> * Synergy Graph Synthesizer:
> The Synergy Graph Synthesizer is designed to reconstruct both node features and graph topology at the client level. With the sophisticated alignment mechanism, OASIS ensures that the synthesized graphs retain structural integrity, closely mimicking the  topology of the original graph. Meanwhile, the synthesis of adjacency matrices can
> provide a means of capturing the most relevant topological relationships in the absence of direct edge-sharing across clients. It potentially helps approximate missing edges.
>
>
> * Topological Codebook:
> The Topological Codebook in OASIS introduces a latent space to represent graph structure at a granular level, capturing subtle neighborhood variations. By quantizing nodes based on their local substructures and encoding them as discrete tokens, we can ensure that even missing edges are represented through these structural tokens. This encoding helps in maintaining the  spatial characteristics of the graph without the need for explicitly shared edge information.
>
>
> In conclusion, the combination of two modules provides a robust mechanism for mitigating the effects of missing edges.  We will update the manuscript to clarify these aspects, emphasizing how our framework can handle edge omission effectively.
>
> **`Limitation 1`: The loss function of the Synergy Graph Synthesizer should be presented in the main text.**
>
> Thank you for your valuable comment. We appreciate your careful reading and for pointing this out. Due to the **strict page limit**, we had to place some technical details in the appendix to ensure the clarity of the main text. In the revision, we will move the description of the loss function into the main body of the paper to improve completeness.
>
> Thank you for your constructive comments. We would greatly appreciate it if you could potentially reconsider the evaluation. Please let us know if further clarification is needed.
>
> Best Regards,
>
> Authors

---

> ### Author Response · Authors · 2025-08-03
> **A Gentle Reminder of Feedbacks**
>
> ***Dear Reviewer Eosq:***
>
> Thank you for your evaluation of our work and the valuable feedback you have provided. As the rebuttal phase is now more than halfway through, we would like to gently follow up and invite further discussion with you regarding our work.
>
> In our previous response, we have carefully addressed the concerns you raised, including the computational complexity of OASIS and its capability to generalize to different GNN models. Moreover, we also clarify the advantages of OASIS over existing baselines and its potential in tackling the challenge of missing edges.
>
> We truly appreciate your engagement, and we also note that other reviewers have recognized the effectiveness and novelty of our work through this rebuttal process.
> We would be grateful for the opportunity to clarify any remaining questions you may have, as we believe this discussion will help us further strengthen our work and its contribution to the field.
>
> Thank you again for your time and consideration.
>
> Best Regards,
>
> Authors

---

> > ### Comment · Area_Chair_BNhx · 2025-08-05
> > **reminder**
> >
> > Dear reviewer,
> >
> > Firstly, thank you for your service! Please take a look at the author's response and engage with the content to maintain or revise your evaluation.  Note that the discussion period has been extended to *Aug 8, 11.59pm AoE*.
> >
> > Thank you,
> >
> > -AC

---

### Official Review · Reviewer_BhNM · 2025-07-03

**Clarity:** 3
**Significance:** 3
**Originality:** 4
**Rating:** 5
**Confidence:** 4

**Summary:**

This paper investigates a novel approach to training Graph Neural Networks (GNNs) within the One-shot Federated Learning (OFL) paradigm. While prior OFL methods have primarily focused on image-based tasks, their direct application to graph-structured data often proves ineffective due to fundamental differences in data characteristics. To address this gap, the authors propose a new framework that synthesizes informative graphs and introduces a topological codebook to better capture graph-specific structural patterns. Furthermore, the framework leverages WESAD and WDSRD mechanisms to enable efficient structural knowledge transfer at the server level. Experiments are conducted to demonstrate the effectiveness of the proposed method.

**Questions:**

Please address my concerns in the weaknesses above.

**Ethical Concerns:**

["NO or VERY MINOR ethics concerns only"]

**Limitations:**

yes

**Quality:**

4

**Strengths And Weaknesses:**

Strengths:
1. This work tackles the novel and challenging task of One-shot Federated Graph Learning (FGL), introducing a dedicated framework tailored to this setting.
2. The paper is well-written with detailed and informative figures (e.g. Figure 2 & 3) to clarify the motivation and the framework.
3. The mathematical formulations are detailed and precise, offering a rigorous description of the model components and training objectives, which facilitates reproducibility and theoretical understanding.
4. The proposed method is rigorously evaluated through comprehensive experiments, benchmarked against a variety of baselines, including conventional FL, existing FGL techniques, and other OFL approaches. Experimental results consistently demonstrate the superior performance of the proposed framework over competing methods.

Weaknesses:
1. The authors need to further clarify the relationship between WESAD and WDSRD on the server side. Why is it necessary to combine these two distillation schemes?
2. Most existing and widely used knowledge distillation methods adopt KL divergence as the loss function. In contrast, this work utilizes a combination of WESAD and WDSRD for knowledge distillation. What are the fundamental differences between these approaches and KL divergence? In other words, for Eq. (10) and Eq. (19), one could theoretically replace them with KL divergence. I hope the authors can explain the distinctions and potential advantages of their proposed distillation strategy over the standard KL-based approach. If possible, including a small comparative experiment would strengthen the argument.

---

> ### Author Rebuttal · Authors · 2025-07-29
>
> ***Dear Reviewer BhNM:***
>
> We sincerely appreciate your insightful review and encouraging evaluation of our work. We are thankful for your acknowledgment of the novelty and significance of OASIS in advancing One-Shot Federated Graph Learning. Below, we provide detailed responses to your comments and questions.
>
>
>
> **`Weakness 1`: The relationship between WESAD and WDSRD needs to be further clarified.**
>
> Thank you for your insightful question. We clarify that WESAD and WDSRD serve two complementary but fundamentally distinct purposes in server-side knowledge integration, and their combination is essential for fully transferring fine-grained knowledge to the global model.
>
> Specifically, WESAD distills semantic knowledge, focusing on the alignment of class-level output distributions between local teacher and global student models. By leveraging a discrete Wasserstein distance informed by semantic affinity, WESAD allows **soft** matching across semantically related categories, which is particularly useful in federated settings where label distributions can vary across clients.
>
> In contrast, WDSRD aligns node-level structural patterns between the teacher and student models by comparing their soft assignments (computed by the distance between node-level representations and topological codebook tokens in eq.(15)). Through the closed-form Wasserstein distance between Gaussian distributions, WDSRD captures both mean-level and covariance-level discrepancies in the assignment distributions, which reflects richer structural variation than simply aligning logits or embeddings.
>
> The combination of WESAD and WDSRD enables a holistic transfer: WESAD ensures that the global model learns to make semantically consistent predictions across diverse clients, while WDSRD ensures that the global model preserves important topological patterns from local graph structures. Our ablation studies in Section 4.3 confirm that omitting either component significantly degrades performance, demonstrating their necessity and complementarity.
>
> **`Weakness 2`: The distinctions and potential advantages of WESAD & WDSRD strategy over the standard KL-based approach.**
>
> Thank you for your concern. We appreciate the chance to clarify:
>
> KL divergence operates under a strict one-to-one alignment assumption, penalizing any discrepancy between predicted probabilities regardless of semantic proximity. This can be suboptimal when semantic overlap exists across categories, which is often the case in heterogeneous clients. In contrast, WESAD introduces a semantic-aware optimal transport (OT) formulation that allows probability mass to flow between semantically similar categories, guided by a learnable affinity matrix. This soft matching provides more flexibility and better generalization, especially under label distribution shift.
>
> For structural knowledge, applying KL divergence directly to the node-to-code assignment distributions would only account for first-order differences and ignore second-order statistics, such as the shape and spread of the structural representations. In contrast, WDSRD models these assignments as Gaussian distributions and uses the closed-form Wasserstein distance, which simultaneously aligns both the means and covariances of local and global distributions. This richer alignment captures the inherent variability and uncertainty in node structure assignments, leading to better preservation of topological information.
>
> To further support our theoretical analysis, we conduct a comparative experiment on Cora, CiteSeer, Coauthor-CS and Roman-Empire datasets. Specifically, we compare our proposed distillation strategy (WESAD & WDSRD) with the standard KL divergence-based distillation. The results are shown in Table 1.
>
> *Table 1: Comparison of Wasserstein-based distillation and KL divergence-based distillation.*
> |Distillation Methods|Cora|CiteSeer	| Coauthor-CS | Roman-Empire |
> |------|------|-------| ---|----|
> |KL Divergence	|48.12	|38.20|59.18 |28.55 |
> |WESAD & WDSRD (ours)|**49.59**|**45.69**| **60.44**| **30.07**|
>
> As shown, our distillation strategy enables more fine-grained and efficient transfer of both semantic and structural knowledge, leading to improved generalization of the global model.
>
>
> Thank you again for your supportive feedback. We hope our explanations address your concerns. If you have any further questions or suggestions, please feel free to let us know.
>
> Best Regards,
>
> Authors

---

> > ### Comment · Reviewer_BhNM · 2025-08-01
> >
> > I have reviewed the responses. The explanations, as well as the comparative experiment, resolve my concern about WESAD and WDSRD. The proposed framework is meaningful and presents a new step in FGL. I will maintain my positive rating.

---

> > > ### Author Response · Authors · 2025-08-02
> > >
> > > ***Dear Reviewer BhNM:***
> > >
> > > Thank you again for your positive feedback and your thoughtful review. Your support means a great deal to us and motivates us to continue advancing in the FGL field.
> > >
> > > Best Regards,
> > >
> > > Authors

---

### Note · Authors · 2025-08-12

***Dear Area Chair,***

We would like to begin by expressing our sincere gratitude for your exceptional leadership throughout the review process. Your commitment to ensuring a thorough and fair evaluation of our manuscript has been greatly appreciated. We are also thankful for the valuable time and attention you have dedicated to overseeing the review.

As for the rebuttal process, we are pleased to report that the majority of participating reviewers have recognized our efforts in addressing their concerns, and have accordingly **voted in favor of accepting our manuscript**:

* `Reviewer BhNM` expressed satisfaction with our detailed explanation and the comparative experiments conducted on the WESAD and WDSRD modules.

* `Reviewer a6gX` retained their positive evaluation following our supplementary experiments addressing more severe non-IID distribution scenarios.

* `Reviewer seeB` indicated a willingness to raise their score after considering our point-by-point clarifications and the additional experiments we presented.

In responding to the concerns raised by `Reviewer Eosq` and `Reviewer 4jz9`, we applied the same level of careful attention and thoroughness:

* For `Reviewer Eosq`, we provided an extensive response to their inquiries concerning the overall computing complexity, theoretical analysis, and the performance of OASIS on other significant GNN models. Additionally, we offered supplementary experiments and theoretical justifications to address each concern raised.

* As for `Reviewer 4jz9`, while they acknowledged our responses, they raised additional questions based on our rebuttal. We have carefully addressed these new queries, presenting further experimental evidence and a more refined theoretical analysis.

However, we remain somewhat uncertain whether our revisions have fully resolved all of their concerns. We hope that the revisions we have made will be found satisfactory.

In light of these developments, we trust that the review process has provided a clear opportunity to address the concerns raised by the reviewers. While we are pleased to have resolved the majority of these issues, we remain hopeful that the efforts we’ve made to refine and strengthen our work will be thoroughly considered in your final evaluation.

Once again, we extend our sincere thanks for your guidance and thoughtful consideration of our manuscript. We look forward to your fair and balanced assessment of our work.

Best regards,

Authors

---

### Decision · Program_Chairs · 2025-09-17

**Decision:**

Accept (poster)

**Comment:**

This work proposes OASIS, a one-shot federated learning framework for graph data. The framework introduces a Synergy Graph Synthesizer and a Topological Codebook to capture structural patterns, along with Wasserstein-based distillation methods (WESAD and WDSRD) to transfer knowledge to a global model.

Reviewers generally leaned positively with only one exception, and evaluated the work favorably.  Authors were able to engage in a fruitful discussion period during which several reviewers improved their scores and maintained positive assessments.

Main feedback:

- several reviewers raised concerns around low baseline accuracy compared to original works, but the authors addressed this by explaining experiments were conducted under high data heterogeneity (low alpha values) (Eosq, 4jz9, seeB)

- several reviewers criticized the lack of theoretical justification for the proposed components, but the authors addressed this by providing a new bound on the information retention of the synthesized graph on global model generalization (Eosq, 4jz9)

- reviewers pointed out scalability concerns re: the Wasserstein-based losses, though the authors clarified that there are efficient approximations to scale up the method and time is saved also by the one-shot design (Eosq, 4jz9).